# Large Language Models' Understanding of Mathematics: Source Criticism and Extrapolation

## Abstract

It has been suggested that large language models such as GPT-4 have acquired some form of understanding beyond the correlations among the words in text including some understanding of mathematics as well. Here, we perform a critical inquiry into this claim by evaluating the mathematical understanding of the GPT-4 model. Considering that GPT-4's training set is a secret, it is not straightforward to evaluate whether the model's correct answers are based on a mathematical understanding or based on replication of proofs that the model has seen before. We specifically craft mathematical questions which their formal proofs are not readily available on the web, proofs that are more likely not seen by the GPT-4. We see that GPT-4 is unable to solve those problems despite their simplicity. It is hard to find scientific evidence suggesting that GPT-4 has acquired an understanding of even basic mathematical concepts. A straightforward way to find failure modes of GPT-4 in theorem proving is to craft questions where their formal proofs are not available on the web. Our finding suggests that GPT-4's ability is to reproduce, rephrase, and polish the mathematical proofs that it has seen before, and not in grasping mathematical concepts. We also see that GPT-4's ability to prove mathematical theorems is continuously expanding over time despite the claim that it is a fixed model. We suggest that the task of proving mathematical theorems in formal language is comparable to the methods used in search engines such as Google while predicting the next word in a sentence may be a misguided approach, a recipe that often leads to excessive extrapolation and eventual failures. Prompting the GPT-4 over and over may benefit the GPT-4 and the OpenAI, but we question whether it is valuable for machine learning or for theorem proving.

## 1  Introduction

There is something perplexing about the proposal set forth by some scientists that large language models (LLM) like OpenAI's GPT-4 have attained some form of understanding that extends beyond mere statistical correlations among words. The day after the release of GPT-4, Ilya Sutskever, OpenAI's chief scientist, openly extols the capacity of this model: "On the surface, it may look like learning correlations in texts, but it turns out that to just learn statistic correlations in text, to compress information really well, what the neural network learns is some representation of the process that produced the text. This text is actually a projection of the world. There is a world out there and this text is the projection of that world. What the neural network is learning is aspects of the world, of people, of the human conditions, their hopes, dreams, motivations, their interactions." In another showcase in GPT-4's technical report, the model shows that it can discern the humor embedded within an image in which a cell phone is connected to a VGA connector (OpenAI, 2023).

When Sutskever elucidates that the product of the text generation process constitutes "the projection of the world" that is "out there.", what lies in the "out there" are composite textual corpus originating from distinct sources, bearing traces of what, in the terminology of "source criticism," could be regarded as the mixture of textual origins. Source criticism, a recognized scientific methodology for dissecting the composition of textual works, can help evaluate claims related to LLM's capacity to comprehend human knowledge. In

particular, the sources encountered by the GPT-4 model are vital for appraising the novel problem-solving capacities it may exhibit.

Recently, some in the research community find value in solving mathematical theorems with GPT-4. Their approach, sometimes, involves repeatedly formulating prompts presented to the model, persisting until the model eventually furnishes the correct solution. It may take for the GPT-4 more than 100 rounds of prompting until it solves some of the problems while for many of the theorems, it will continue to fail. Nevertheless, when GPT-4 succeeds in solving a few of the theorems in datasets such as miniF2F (Zheng et al., 2021), its success may be reported in papers to indicate scientific progress. A progress that tends to be an occasion for signaling machine intelligence - that machines have developed the crucial capability of following mathematical logic and solving formal theorems.

The question that arises here is whether the GPT-4 model is understanding any of the mathematical concepts. If we assume, as Sutskever suggests, that text is the projection of the world and the process that produces the text, i.e., the human knowledge, is there any evidence that GPT-4 has grasped some of the mathematical knowledge from the mathematical text? Is GPT-4 merely repeating the proofs that it has already been exposed to, or does it have some understanding of the mathematics that it deploys when it provides the correct proofs? To answer this question, here, we perform a scientific inquiry on the mathematical understanding of GPT-4. One can consider this inquiry as a case study to understand the model better seeking guidance on how to think about its capabilities and how to use it. With that understanding, we might rethink the usefulness of prompting the GPT-4 over and over again in order to solve a mathematical theorem. Our results suggest that the task of theorem proving is more of a retrieval task such as Google's search engine (Brin & Page, 1998) as opposed to the task of predicting the next word in a sentence.

## 2 Source criticism

Source criticism is a known method in scientific inquiry, research methodology, information science, epistemology, and many other fields (Hjørland, 2012). In short, it refers to evaluating an information source. Specifically, we would like to know the sources that the GPT-4 model has likely seen. With that knowledge, we may be able to evaluate this model's ability in solving mathematical theorems that it has not seen before.

While the training set of GPT-4 is kept as a secret, we can make some inferences about it:

1. Creators of some adjacent models have been transparent about the training set of their models. For example, the GPT-f model was trained on the Common Crawl and WebMath datasets (Polu et al., 2022). The WebMath contains all the publicly available code and text on GitHub, arXiv, and Math StackExchange.

2. Any high quality text that is readily available on the web, anything that may be grabbed by the Common Crawl or any other web scrapping method, may have been used in the training of GPT-4. This could include any theorems in the Mathlib library (Moura & Ullrich, 2021), any proof that can be grabbed from the public, or private domain, or be created by experts specifically for training of the model.

3. Numerous researchers are interacting with the GPT models, sometimes on a daily basis, crafting various prompts/questions/statements. These prompts may be subsequently used as a training source for GPT. For example, consider that a person manually writes the proof for a theorem in the miniF2F dataset, a theorem that GPT cannot solve yet. That person might feed that proof to the GPT, perhaps asking the GPT to polish its writing. That proof can then be picked by the GPT model, and later on, it might be able to prove that theorem for any user. Note that although the parameters of the model may not be updated frequently, it's outputs are dependent on its prompt history and the text it has seen so far. It is common knowledge that the outputs of GPT models change and improve over time (Chen et al., 2023). We provide more evidence on the evolution of GPT-4 later in this section.

The extent of information on the Common Crawl is particularly interesting. The GPT-f model developed for mathematical learning was trained on 300 billion tokens from CommonCrawl. The GPT-3 model was trained on an enhanced version of Common Crawl as well as other sources such as Wikipedia (Brown et al., 2020). The size of the most recent CommonCrawl is 390 TiB including the contents of 3.1 billion pages on the web. Hence, it may be safe to assume that any formal mathematical proof that is available on the web may have been used as a training source for GPT-4. As pointed out by Reichel et al. (2023), any proof that is released maybe used as data for new machine learning models.

### 2.1 How forthcoming is GPT-4 about its sources and its capabilities?

Clearly, the interactions that researchers have with GPT-4 is beneficial for the model and for OpenAI. The mere perception that GPT-4 has generalization capabilities adds to the appeal for scientists and researchers to keep interacting with it to solve mathematical theorems. If a student or a researcher adopts the view that GPT-4 is a fixed model with understanding of mathematics and generalization capabilities, and the only way to change the outputs of GPT-4 is to provide better prompts, then they might dedicate a considerable effort to keep prompting the model to solve a specific theorem, and when the model, eventually, solves that theorem, they might consider it as progress in generalization capabilities of AI. One might think that it is because of the efforts put into crafting the prompt that the GPT-4 was able to solve the theorem, because GPT-4 is a fixed model.

GPT-4 itself reinforces this perception. When we asked about its training set, mathematical proofs in its training set, and its latest updates, the response was as following

> **GPT-4:** *My last training data goes up until January 2022. … I don't automatically update with new data. You're interacting with a fixed version of me as of that 2022 cut-off…*
>
> *My knowledge, including that of Lean mathematical proofs, is based on data available up to January 2022. I was trained on a mixture of licensed data, data created by human trainers, and publicly available data. However, I do not have real-time updates, so any developments, proofs, or advancements in the Lean community or other fields after January 2022 would not be known to me…*
>
> *I don't have access to my training data, but I was trained on a mixture of licensed data, data created by human trainers, and publicly available data. OpenAI, the organization behind Assistant, has not publicly disclosed the specifics of the individual datasets used, including whether any specific data source was included…*
>
> *OpenAI hasn't released detailed specifics about individual datasets, including whether any specific dataset related to Lean mathematical proofs was used. However, I was trained on a diverse range of internet text. If there are publicly available materials, tutorials, discussions, or other text-based content on Lean mathematical proofs up until my last training cut-off in January 2022, then it's likely that I've been exposed to the general concepts, methodologies, and techniques.*

Researchers in the field refer to this claim to emphasize the importance of their success in proving theorems with GPT-4. To see our questions and GPT-4's complete responses, please refer to Appendix A.

### 2.2 Evidence for continual improvement of GPT-4 and the expansion of its information sources

Let us see if there is evidence contradicting the claim that GPT-4 is a fixed model with fixed information sources. Let's consider one example. In June 2023, when we asked GPT-4 to prove the following theorem in lean, its response was that it cannot solve this theorem and its proof may require "a great deal of effort", and it did not provide any form of proof. Here is the theorem and part of GPT-4's response:

> **User:** *please prove this in lean: lemma $(t : \mathbb{R}) : ((2^t - 3 * t) * t)/(4^t) \leq 1/12 :=$*

> **GPT-4:** *For inequalities like this one, calculus methods like taking derivatives to find critical points are often helpful, but Lean's real number library doesn't include comprehensive calculus support out-of-the-box as of my last update in September 2021.*
>
> *....*
>
> *It's also possible that this specific inequality would require advanced mathematical methods to prove, and Lean's real number library might not be sufficiently developed to handle them without a great deal of effort.*

However, when in October 2023, we posed the exact same question to GPT-4, it immediately provided a partial proof for it.

Let us look at a second example from June 2023. When we asked GPT-4 to prove this:

$$theorem\ (x\ y : \mathbb{R})(h : x \neq y) : 0 < (x - y)^2 :=,$$

it provided a wrong proof using the *pow_pos* lemma from the mathlib library. The *pow_pos* lemma states that power $n : \mathbb{N}$ of a positive number is also positive. Hence, *pow_pos* can only prove $0 < (x - y)^2$, if we have $0 < x - y$. But, the theorem statement given to GPT-4 does not include such positivity statement, so *pow_pos* is not applicable here. The correct proof should use the fact that the square of any real number is non-negative, and then eliminate the possibility that $(x - y)^2$ can be equal to zero, and reach the goal based on those steps.

In June 2023, when we asked GPT-4 to prove a similar alteration of the same theorem:

$$theorem\ (x\ y : \mathbb{R})(h : (x - y) \neq 0) : 0 < (x - y)^2 :=,$$

it used the same incorrect approach to prove it using *pow_pos*.

However, when in October 2023, we asked the exact same question, GPT-4 provided the correct proof on the first prompt.

The two examples above seem to be clear indications that GPT-4's information sources and capabilities are continuously expanding. But, how can we reconcile this with the GPT-4's claim that it is a fixed model and it has not been updated since January 2022? Perhaps GPT-4 is not continually going through training updates, but it might be possible that GPT-4 consists of modules that look through additional sources and when there is a match between the prompt and those sources, they are utilized to produce its responses. It is clear that GPT-4 is not just a single module model that predicts the next word, but on top of that, there is a sophisticated system of modules that pre-process the prompts and regulate its outputs.

A person who interacts with GPT-4, in October 2023, might attribute the correct proofs to the generalization capabilities of GPT-4. However, when we consider the GPT-4's inability to solve the same theorems, 4 months before, we shall reconsider the claim that GPT-4 is a fixed model. We may also reconsider whether solving such theorems with GPT-4 is progress and whether it is worthwhile to report to the research community that one has solved such theorem with GPT-4.

If we adopt the view that GPT-4's sources of information is routinely updated and expanded based on the prompts and other information sources that gradually become available, then, when GPT-4 eventually solves a mathematical theorem, its success may be attributed to the possibility that somehow the correct proof has made its way to the information sources of the model. Perhaps someone on one corner of the globe, has provided the proof to the model, and now the model is reproducing that proof for everyone else who is asking for that proof.

## 3 Evaluating the understanding of the model

There are standard educational assessment methods designed to evaluate various degrees of understanding of learners. In the era where text books come with their solution manuals, there are still ways to assess the knowledge of a student on specific topics.

Interesting evaluation techniques have been proposed to evaluate mathematical reasoning abilities of models such as GPT-4. For example, Liu et al. (2023) propose a method to evaluate whether a model is capable

of identifying misconceptions in mathematical reasoning. They also prompt the model to make certain inferences based on a given misconception. These are interesting approaches in educational studies which can be insightful and sophisticated. In another approach, Wu et al. (2023) evaluate the GPT-4's abilities through counterfactual tasks and their results suggests that GPT-4 is mostly good at reciting and it is not good at reasoning. Yiu et al. (2023), also, finds LLMs to be imitation engines and reports lack of innovation capabilities in them compared to human children. Srivastava et al. (2023) provide a self-called "extremely difficult and diverse benchmark" for a variety of tasks including mathematics with the goal to measure the capabilities of large language models. Clearly, existing models, including GPT-4 do not do well on this benchmark.

In summary, there are abundant difficult problems that GPT-4 cannot solve. The inabilities of GPT-4 includes a range of problems from math Olympiad problems to very simple problems. At the same time, there are a wide range of problems that GPT-4 can solve accurately. What is not clear is how we can interpret the successes and failures of GPT-4. How much of its correct answers can be attributed to its generalization capabilities, and how much of them are merely replication of correct answers from its training set?

Given the secrecy of its training set, which is a deliberate choice by the owners of GPT-4, it is not straightforward to find the answer. As mentioned above, many have tried to evaluate this model on harder and harder problems. This approach removes the fog regarding the exaggerated capabilities of GPT-4 and LLMs, but many may still focus on the correct answers of GPT-4 and interpret them as its generalization capabilities. Most recently, Yu et al. (2023) proposed an evaluation method, called SKILL-MIX, in which prompts are designed to include a combination of skills such as metaphor, red herring, and common knowledge physics. The combination of such skills in the prompts would entail that a sensible answer, most likely, would not exist exactly in the training set of the model, and for the model to produce a sensible answer, it will need to draw from various parts of its training set. The results indicate some positive evidence that the GPT-4 can sometimes provide sensible answers to prompts combining a small of number of those skills, but for mst of the prompts, it does not succeed. It is notable, however, that GPT-4 performs considerably better than other LLMs. While we find this evaluation technique insightful, we note that it still aims at evaluating the model on more complex and harder questions.

For mathematical reasoning, however, we believe that the evaluation of generalization capabilities of LLMs can focus on an approach examining the foundations. We do not look for hard math Olympiad challenges. We do not aim to craft questions combining elements from number theory, algebra, and geometry, altogether. Rather, we look for relatively simple and trivial questions. Our only criteria is to pose questions where the formal proof is not readily available on the web. To verify whether the proof for a lean theorem is available online or not, we use the Google search engine.

Moreover, since our goal is to evaluate the mathematical understanding of the GPT-4 model, we directly give the problem statement in formal Lean language as opposed to the natural language. This way, possible mistakes of translating to formal language will not arise.

### 3.1 Case 1

A natural number is either 0, 1, 2, 3, 4, 5, 6, or it is greater than 6.

$$lemma\ (x : \mathbb{N}) : x = 0 \vee x = 1 \vee x = 2 \vee x = 3 \vee x = 4 \vee x = 5 \vee x = 6 \vee 6 < x :=$$

GPT-4's proof, provided in the Appendix A, uses an imaginary term *nat.cases_on* without defining it. *nat.cases_on* is commonly used in the lean community, but it needs to be defined as a function before being used. Another distinct feature in the GPT-4's proof for this lemma is the use of the term *or.inl rfl*. We googled both of these terms together, leading to only 6 results from the entire web. All the six results are official lean documents/files. The instance where *cases_on* is used (Avigad et al., 2021, p. 96) corresponds to defining an enumerated type for the days of a week: 1, 2, 3, 4, 5, 6, 7, in appearance, very similar to what we have in the statement of our lemma. In those same documents, we see the instance of *or.inl rfl* as well.

One could argue that GPT-4 has found something very similar to the statement in the prompt which is a good capability, albeit not useful for proving our lemma. If we were to define an enumerated type for

numbers 0 to 6, of course, we can use it to prove x can be either of them. However, it would not help with proving our lemma as the only statement given in the lemma is the definition of natural numbers. This suggests that GPT-4 might have interpolated to the closest match between its training set and the statement of the lemma, and it does not have an understanding of the context. One could argue that if GPT-4 had seen the proof for our lemma, it would have interpolated to that proof instead because it would have been more similar to our prompt.

### 3.2 Case 2

We modify the previous lemma turning it into the following form:

$$lemma \ (x : \mathbb{N}) : x < 5 \lor x = 5 \lor x = 6 \lor x = 7 \lor x = 8 \lor x = 9 \lor 9 < x :=$$

This time, GPT-4 did not use its previous approach, i.e., it did not map to the enumeration type example of weekdays in the lean library. Instead, it tried to use *cases* in conjunction with *nat.zero_lt_succ* from the mathlib library. The *nat.zero_lt_succ* states that if we add 1 to any natural number, its results would be greater than 0. Clearly, this approach would not solve the problem.

We googled for the term *exact nat.zero_lt_succ* used in GPT-4's proof. Google returned only 4 results, all of them were official lean proofs written by experts. Of these four proofs, two of them were related to harmonic series, and one of them related to power series. Again, it seems that GPT-4 did not have a good point of reference, and it just picked an irrelevant lemma that it had seen being used in similar situations. When we googled for the term *exact zero_lt_succ*, there was no exact matches.

### 3.3 Case 3

The square of no natural number can be 27. Here is the prompt we gave to GPT-4:

$$please \ prove \ this \ in \ lean: \ (x : \mathbb{N})(h : x^2 = 27) \to false$$

The proof provided by GPT-4, as shown in the Appendix A, is incorrect. The GPT-4's proof first defines a new variable $y = \sqrt{27}$ as a natural number. Then it aims to prove that $y^2 < 27$. Obviously it fails to prove this. Then it aims to prove that $y < x \land x < y + 1 + 1$. The GPT-4's approach goes nowhere near proving the theorem.

We made the problem simpler by adding an additional statement making $6 \leq x$:

$$(x : \mathbb{N})(h_1 : x^2 = 27)(h_2 : 6 \leq x) \to false$$

This can be proved even for real numbers. Since we have $6 \leq x$, it is easy to prove that $x^2$ cannot be equal to 27. However, GPT-4 still struggles with proving this simplified lemma. This time, GPT-4 starts by proving that $36 \leq x^2$. Then it replaces $x^2$ with 27 to obtain $36 \leq 27$. This can be used as a contradiction, however GPT-4 proceeds with using the *not_le_of_lt* lemma. This lemma cannot lead to our false goal, because it states that if $a < b$, then $\neg b \geq a$. So, even in such an overly simplified case, although GPT-4 made some progress, it could not provide the correct proof.

### 3.4 Case 4

If a, b, c, x, y, and z are positive real numbers where $c \leq b \leq a$ and $x \leq y \leq z$, then we have $x/c + y/a + z/b \leq x/a + y/b + z/c$.

$$please \ prove \ this \ in \ lean:$$

$$lemma \ (a \ b \ c : \mathbb{R})(x \ y \ z : \mathbb{R})(h_0 : 0 < a \ \land \ 0 < b \ \land \ 0 < c)$$

$$(h_1 : 0 < x \ \land \ 0 < y \ \land \ 0 < z)(h_2 : c \leq b \ \land \ b \leq a)(h_3 : x \leq y \ \land \ y \leq z) :$$

$$x/c + y/a + z/b \leq x/a + y/b + z/c :=$$

Again, GPT-4 does not take a sensible approach. It first aims to prove that $x/c \leq x/a$ which is obviously false, and it fails to prove it. The GPT-4's overall approach to this problem was naive, as it aimed to prove $x/c \leq x/a$, then $y/a \leq y/b$, and $z/b \leq z/c$, planning to eventually add up all the three inequalities to obtain the goal.

A student may also think of taking such an approach, but soon, they may realize that $x/c \leq x/a$ is not true and such an approach would not work, but GPT-4 does not realize that.

### 3.5 Case 5

Our last case is based on the property that squaring both sides of an inequality reverses the inequality when both sides are negative real numbers:

$$please \text{ prove this in lean:}$$

$$lemma \ (x : \mathbb{R})(h_0 : 0 \leq 1 + 2x)$$

$$(h_1 : -11x - 9 < -(2x + 9)\sqrt{1 + 2x})$$

$$(h_2 : -(2x + 9)\sqrt{1 + 2x} \leq 0) :$$

$$(-11x - 9)^2 > (-(2x + 9)\sqrt{1 + 2 * x})^2 :=$$

In $h_1$, we have an inequality where both sides are negative. The goal of the lemma states that when we square both sides of this inequality, the direction of the inequality sign flips. Lean library already has the lemma for flipping the direction of inequalities. The theorem statement also gives the information that the right hand side of inequality is negative. It follows that the left hand side is also negative. The lean proof can be written in one line.

However, when we ask GPT-4 to prove this, it takes a completely misguided approach. It first aims to prove that $-(2x + 9) \geq 0$ which is untrue and it fails to prove it. Then, it proceeds with squaring both sides of $h_1$. In the comments of its proof, GPT-4 states: "Now, we can square both sides of the inequality and maintain the direction of the inequality. This is due to both sides being non-positive." This statement by itself is wrong regardless of its applicability to our lemma.

Google search did not return any results including both of the lemmas used in GPT-4's proof. This suggests that GPT-4 came up with this novel and wrong answer by itself.

If we assume that GPT-4 has some understanding of basic mathematical concepts, proving this lemma does not require extrapolation beyond its familiar concepts, the concepts that are used in its correct answers. However, if we consider the lean statements merely as a text corpus, then the theorem posed in this case may be quite novel, as its formal proof is not available on the web, and apparently, not seen by the model. From such a textual perspective, this theorem would require considerable extrapolation, which GPT-4 tries to perform, leading to a failure.

### 3.6 Our take from these cases

Our only criteria for choosing these lemmas was the lack of availability of a formal lean proof for them in the public domain. These 5 cases are selected, as a show case, from a larger pool of easy problems that GPT-4 could not solve. The problems that we chose for this showcase were intentionally all about natural and real numbers. We did not present cases where GPT-4 fails to solve quadratic equations, or cases related to rational numbers. We did not go for Olympiad problems or extremely hard cases. The easy way to find a failure mode of GPT-4 is to rely on Google's search engine. We have to look for problems that do not have a formal proof readily available online. To evaluate a learner's understanding of a subject, the evaluator has to first cover the basics.

# 4 Ways to view and use the GPT-4 when solving math theorems

## 4.1 The evidence for success and failure of GPT-4

We have seen that, currently, GPT-4 is the best available tool for replicating/rephrasing/polishing existing text. The same capability seems to be true for mathematical proofs. GPT-4 is able to provide the correct proof for a variety of mathematical theorems, and the correct proof is generated time after time, with consistency, and insensitive to the minute details of the prompt. Such proofs appear to be the ones that already exist in a formal language in the public domain, the proofs that are likely used in the training process of GPT-4.

If we adopt this notion that GPT-4 is good at providing the proof for problems that it has seen in its training process, that could give us a clear guideline on how to deploy this model and how to think about it. The developers and owners of the GPT-4 certainly have access to the training set of their model. They can identify the theorems and the proofs that the model is trained on. They can measure how accurate the model is in solving theorems that are not included in its training set. They can also create procedures for the model to abstain from attempting to prove theorems that it is unlikely to prove correctly. The approach of OpenAI, so far, seem to be encouraging the community to keep prompting the GPT-4 while constantly improving the model behind the curtains, perhaps using the same prompts they receive from the community, i.e., creating a fog about generalization capabilities of their model while constantly improving it.

If the success rate of the model in proving unseen mathematical theorems, the like of which we explored in this paper, is minute, then it would be useful to report that to the community at the least. Such transparency would benefit the community as a whole, and it may benefit the OpenAI as well in the long run.

## 4.2 Training the models on all the existing proofs

The practice of grabbing all the available proofs and feeding it to the model could be fine if we are using the model as a writing assistant for replicating/rephrasing/polishing existing proofs. But, when we are not upfront about the model's generalization capabilities, that practice may turn out to be misleading for the broader community. For example, if we work based on the idea that GPT-4 is only good at reproducing the proofs that it has seen before, and it is not a tool for solving novel mathematical theorems, then we would not assign value to prompting the model day after day to solve a trivial theorem. Instead of encouraging a PhD student to keep prompting the GPT-4 until they can get the correct proof for a mathematical theorem, we may encourage the same student to write the proof themselves, if that problem needs to be solved. After, we have the proof, one can still feed that proof to the GPT-4, or other LLMs, so that the model can polish it and reproduce it for anyone around the globe who needs that proof, perhaps for educational purposes.

On the other hand, if we need to develop AI models that can solve novel mathematical theorems, i.e., theorems that the AI model has not seen before, we need to have clear procedures for evaluating their generalization capabilities. In that scenario, solving mathematical theorems with GPT-4, when we do not know what the model has seen before, does not appear to be progress towards our goal. Prompting the GPT-4 over and over may improve the capabilities of GPT-4, and the prompts may provide new training source for OpenAI. But, it would not be clear whether the improvement counts as developing an AI model that has learned to solve mathematics. Such progress may well be categorized as developing a model that is good at replicating/rephrasing/polishing existing text made by humans.

# 5 A retrieval task or predicting the next word?

There may be many motivations to use AI for solving mathematical theorems. Automation is one of them. Automation in this context may be used for software engineering purposes. Moreover, computers may be able to perform certain tasks better and faster than humans, especially, the tasks that involve retrieval from a large database, extensive search, or evaluating enormous possibilities. The memory that a computer deploys may be much larger than the memory of a human brain. Most would agree on the usefulness of search engines such as Google which can bring us useful results in a fraction of a second, a task that would

be hard to perform without an automated system. Computers can also outperform humans in certain games such as Chess and Go (Silver et al., 2017).

But imagine we were using the Google's search engine to find links related to a specific topic, and the results came after an hour, and many of the links were broken or irrelevant. Then, you might reconsider the usefulness of this search engine. Using AI to solve mathematical theorems can be seen from a similar perspective. If we solve a mathematical theorem with a large language model when it takes a long time[1] for it to come up with an answer, and that given answer is most likely incorrect, what is the value in that model? GPT-4 predicts the next word, and it does not take long for it to produce a correct or incorrect answer. However, other types of models such as GPT-f rely on extensive trial and error. So, the familiar trade-off between compute expense vs the correctness of the outputs is present here. But it does not seem clear whether using a large language model is the best approach for solving mathematical theorems. Being able to solve unseen mathematical theorems from scratch requires the use of familiar contents, e.g., the lemmas that are available in a library. From this perspective, solving the theorems requires the ability to identify the applicable theorems and lemmas that are available in a very large library. This is similar to the retrieval and page rank method used in the Google's search engine (Langville & Meyer, 2006), and not the method of predicting the next word deployed in large language models.

When we want to find web pages related to a topic, we do not usually ask a language model, instead we use a search engine to retrieve, in real time, from the database of all the web pages. It is more efficient to search the web in real-time than to train a model, and then ask the model to give us the answer. Search and retrieval from a factual database appears to be a fundamentally different task compared to predicting the most plausible next word in a sentence. Suggesting links to web pages is exactly the shortcoming of the large language models. The links that LLMs provide are often non-existent, i.e., hallucinations of the model from the contents of the web. Although these hallucinations are being filtered out more and more every day, it does not fully conceal that the method of predicting the next word can lead to extensive hallucinations.

The same logic applies for retrieving the applicable lemmas and tactics for solving mathematical theorems. We have a library like Mathlib. We would like to find the applicable lemmas, tactics, etc. that would eventually compose the proof for a given theorem. It would be better to go directly to the library, and use a powerful search engine to retrieve the applicable premises in that library and rank them based on their relevance/usefulness. This is the approach that was originally taken by the math community, leading to methods such as Sledgehammer (Böhme & Nipkow, 2010; Paulsson & Blanchette, 2012), but much more work can be done in that direction. Using a language model may not be as useful, for the same reason that we are still using search engines such as Google and Bing to scan the contents of the web. These search engines, first scan the contents of the web, identify the contents relevant to the inquiry, and then rank those contents based on their relevance, and return the ranked list. Image search engines perform a similar procedure. Clearly, powerful search engines also make use of machine learning methods, especially for ranking the results. We view that approach fruitful as well. The point is that relying on a language model that is trained to predict the next word may not be a good approach to generate proofs for mathematical theorems unless those proofs are already written and we just want the language model to reproduce them for others.

Some researchers have considered the idea of using language models in conjunction with premise selection methods such as Sledgehammer. For example, Jiang et al. (2022) uses the Sledgehammer for premise selection and then relies on a language model to ultimately generate the proof for the theorems using the selected premises. This approach is more successful in solving some of the simpler theorems and its computational cost is also less expensive compared to the methods that solely rely on language models. Nevertheless, it still cannot prove more than 70% of the theorems in the miniF2F testing set. While the authors of that work describe the reason for their improvement gain to be the hybridization of various tools, it is also plausible to see their method as a move away from the use of language models for proving novel mathematical theorems.

GPT models may still be useful, and training them on mathematical proofs may still be a good thing to do, if some of us want to use these models as writing assistants to reproduce existing proofs. These models

---

[1]See, for example, the running time of GPT-f model (Polu et al., 2022, Section 7) as well as the running time of GPT-4 when it has to be prompted so many times.

may become more credible if they stick to what they have seen in their training set, reproducing the proofs that are previously verified. This would entail, for the GPT-4 model, to abstain from producing proofs of its own, until and unless it can do that with some verified degree of correctness. The same argument goes for producing web pages and book references. To scan the web or to search for references, using a search engine is faster and more reliable than using GPT-4, a non-retrieval based model. While many of the large language models were initially unrestrained about providing fictional web links and book references, their creators have now implemented strict restraints on them to stick to the exact links and references that they have seen.

## 6  Is it all about compression? What is the role of extrapolation?

Sometimes, the generalization of the GPT models is attributed to the notion of compression, specifically in the context of Kolmogorov complexity (Kolmogorov, 1963). We find the notion of compression and Kolmogorov complexity relevant and insightful in this context, but perhaps this notion has been overemphasized, carrying the burden of much speculations[2] with limited ways for verification. The Kolmogorov complexity of object $x$ with respect to the specifying method $\phi$ is defined as

$$K_\phi(x) = \min_p \{|p| : \phi(p) = x\},$$

and $K_\phi(x) = \infty$ if there are no such $p$ (Li & Vitányi, 2019). Here, $p$ denotes a program describing the $x$ via a method $\phi$, and $|.|$ is the length operator. Hence, the Kolmogorov complexity of an object (e.g., text) is the length of the shortest program that can produce $x$ as output using the programming method $\phi$.

When one creates a model that is simpler than another model yet it can learn better, that aligns with the notion of Kolmogorov complexity, as the simpler model would have a lower Kolmogorov complexity. However, Kolmogorov complexity does not provide a method on how the minimum length program, i.e., compression of the data, should be computed, nor it cannot be used to verify whether a given compression of the data is its best compression (Vereshchagin & Vitányi, 2004). In practice, Kolmogorov complexity is often used to prove impossibilities in algorithmic computations. Another result derived from Kolmogorov complexity is that no program can compute the Kolmogorov complexity for infinitely many elements (Li & Vitányi, 2019), e.g., strings of text. Kolmogorov complexity is indeed not computable. Hence, many of those speculations about generalization of deep learning models, and their ability to acquire an understanding of "the world out there", remain non-verifiable through the lens of compression. However, in the case studies that we presented earlier, GPT-4 has not grasped or compressed an understanding of basic mathematical concepts. The theorems that GPT-4 can prove correctly, proofs seem to be already available on the internet.

To study the generalization, performance, and failure modes of GPT-4 and other deep learning models, it would be better to adopt a broader view and note that compression is not the only operation performed by such models, and as a result, compression, alone, may not be able to explain those phenomenon. To arrive at such broader view, we can look back, even further, on earlier work of Andrey Kolmogorov, not on compression, but on extrapolation (Kolmogorov, 1941), the topic that was pursued independently by Norbert Wiener around the same time.

Kolmogorov is famous for his work on computational complexity while that piece of his work was preceded by his study of extrapolation. Similarly, Wiener's famous work, Cybernetics (Wiener, 1948), was preceded by his work on extrapolation. Indeed, Wieners' work on extrapolation was about sequences in their general form including sequences of words (Wiener, 1942, p. 2). The type of sequence that Wiener originally worked on was about the sequence of locations that a guided missile goes through in space. After the World War II, he extended those same concepts for modeling other sequences such as sequences of words (Cao, 2023).

Many may agree that the ideas put forth in the Wiener's Cybernetics are a foundation for modern methods of learning from the data (Ma et al., 2022), yet the Cybernetics itself was built on the Wiener's work on extrapolation. Both Kolmogorov and Wiener where developing models that can learn a phenomenon from a data distribution, and then, extrapolate from it.

---

[2]See for examples, the talk by Ilya Sutskever at the Simons Institute in August 2023: `https://simons.berkeley.edu/talks/ilya-sutskever-openai-2023-08-14`, and his interview with NVIDIA's CEO on the release of GPT-4.

### 6.1 Does GPT-4 extrapolate?

The answer is yes, in many different ways and to many different extents. When the model rephrases a piece of text from the received prompt or from its training set, that may entail some degree of extrapolation, but the extent of extrapolation may be very small. On the other hand, when the model hallucinates, or when it attempts to write a poem, the extent of extrapolation may be much larger.

For solving mathematical theorems, when the model has seen the proof, it can interpolate to that seen proof and provide the sensible answer. Along the way, it may polish the proof from its training set, or make some small modifications to it. Such modifications may entail small degrees of extrapolation in the textual space.

But, when the model is tasked to generate a proof on a concept that it has not seen before, it may need to extrapolate to a larger extent, on a conceptual, as well as, a textual level. As we saw in our experiments, the model may end up assembling a nonsensical answer to unseen theorems even when the concepts used in those theorems are familiar. So, the extrapolation in the textual space is influential as well. Extrapolation in the text space can be contrasted to extrapolation in the compressed space that a model has learned. But failures of GPT-4 in the cases we investigated in this paper are explainable not by lack of exposure to certain mathematical concepts, rather by unavailability of those proofs in formal language. Those failures where not cases of conceptual extrapolation, but of textual extrapolation.

The extent of extrapolation may be used as a guide to prevent the nonsensical answers and to tie the answers of the model to its training set. This would also be useful to prevent hallucination. Limiting the answers of the model to its training set can be seen as a step towards grounding its answers in verified data. Clearly, the training set itself should be purged from any incorrect text/information.

Of course, if one wants the model to come up with a poem, or to write some fictional statement, excessive extrapolation may be very desirable. But in many other instances, it may be more sensible if the model abstains from extensive extrapolations and acknowledges its limitations.

### 6.2 GPT-4's internal instructions on extrapolation

GPT-4 is not unfamiliar with these notions. When asked whether it extrapolates or not, it responded:

> **GPT-4:** *Extrapolation in the context of ChatGPT refers to generating responses that go beyond simply repeating what is in the training data, aiming to make inferences, generate creative text, or provide explanations based on patterns it has learned.*
> *…*
> *So, in summary, I extrapolate by generating text based on the context of the prompt and the patterns I've learned during training. This allows me to answer a wide range of queries, even those that were not explicitly present in my training data.*

When asked how it avoids excessive extrapolation, it responded:

> **GPT-4:** *Avoiding excessive extrapolation is crucial for maintaining intellectual rigor, whether you're engaged in scientific research, data analysis, or everyday decision-making.*

Hence, unlimited extrapolation is considered undesirable by GPT-4, but it is not clear how much extrapolation is considered to be excessive by the model. This question may be hard to quantify in various domains, e.g., math, literature, etc. Nevertheless, it is notable that GPT-4 acknowledges that it tries to avoid excessive extrapolation.

When we asked GPT-4 if it avoids extrapolation, it replied:

> **GPT-4:** *I don't "avoid" extrapolation in the sense that I actively detect and reject it. Rather, my design inherently minimizes extrapolation by virtue of how I generate responses.*

The word "inherent", in GPT-4's response, appears to be a clear overstatement, considering the cases that we studied in this paper. While GPT-4 abstains from providing a proof for some theorems, those theorems seem to be the ones that are more on the difficult side, and there are not similar instances of such proofs available on the web. Nevertheless, there are a wide variety of theorems, as in the cases we studied in this

paper, where GPT-4 goes off the rail by generating proofs that could be considered unjustified or out-of-context extrapolation. Perhaps this type of extrapolation is not limited or minimized by GPT-4, inherently or otherwise. It would not be surprising if GPT-4 is allowed to go beyond the boundaries of its knowledge, and sometimes give out nonsensical answers. While expert users are constantly prompting the model, giving nonsensical answers could still be useful by the model, as it allows its limitations to be examined by its users, eventually leading to the improvement of the model and expansion of its capabilities. This is definitely a progress towards having a better automated writing assistant.

On the other hand, when asked how it avoids replicating training data and plagiarism, it replied:

> **GPT-4:** *I am designed to avoid generating text that is identical to the training data...*

Again, the answer is related to extrapolation, but this time, GPT-4 is designed to perform some intentional and limited amount of extrapolation in the textual space to ensure that its responses are not identical to its training set. This may be interpreted as no extrapolation on a conceptual level, rather, a simple rephrasing in the textual space may be sufficient to achieve that goal. In other words, it seems that GPT-4 is designed to maintain some minimal level of extrapolation in the textual space in most of its responses.

Looking back at the pioneering work of Kolmogorov and Wiener in the 1940s, the kind of extrapolation that they formulated and pursued was limited, guided, and in specific ways related to specific data distributions. It appears that much more work has to be done to study the extrapolations performed by deep learning models, and at the same time, to limit and guide the extrapolations that they perform. Extrapolation still is an overlooked and under studied concept in the deep learning literature and practice.

## 7 Conclusions

We performed a critical inquiry into the claims about GPT-4's ability to understand mathematics. We specifically posed questions that are simple, but their formal proofs are not abundant on the internet. We observed that GPT-4 was not able to answer those questions correctly. Scientific evidence that GPT-4 has acquired an understanding of the mathematical concepts appear to be scarce. Instead, our results suggests that GPT-4 has the capability to provide correct answers when it has seen the solution before indicating the usefulness of this model in reproducing/rephrasing/polishing existing proofs as opposed to solving mathematical theorems based on mathematical understanding. This may provide us with some guidance on how to use this model, and how to interpret its correct and incorrect answers.

We further discussed the task of theorem proving and suggested that it is more of a retrieval task from a database, similar to how search engines work. We discussed why most people still use retrieval-based search engines to find contents on the web as opposed using the likes of GPT-4. Building powerful search engines for mathematical libraries could be a more fruitful approach compared to training GPT-4 models on all the available formal proofs in the world. We further discussed that this approach of training a model on all contents of the web blurs the line on measuring the generalization capabilities, a line that can be easily de-blurred by creators of GPT-4.

On the other hand, we observed that GPT-4's ability to reproduce the contents of its training set may provide a straightforward approach for it to measure its own confidence. Whenever the model has to go beyond its training contents, at least when it comes to mathematical proofs, it appears that it does not have a reasonable chance of providing sensible answers. It would be useful for the research community if OpenAI honestly reveals the generalization capabilities of its model in solving formal mathematical problems, the capabilities that can be verified only through the knowledge of its training set. We further discussed how extrapolation, in the textual space as well as in the representation space, may be a useful concept to evaluate and identify the success and failure cases of large language models.

Another observation is that mathematical theorem datasets such as miniF2F were created by people at the OpenAI. Putting out a model that outperforms all other models while not being forthcoming about the model's procedures and its training sources, blurring the line to evaluate its generalization capabilities, and at the same time, encouraging the scientific community to keep prompting the privately owned GPT-4 model with the goal to improve it, may seem a bit deceiving and self-serving, and at the same time, counter-productive for scientific progress.

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

# A  Interactions with GPT-4

### Section 2.1

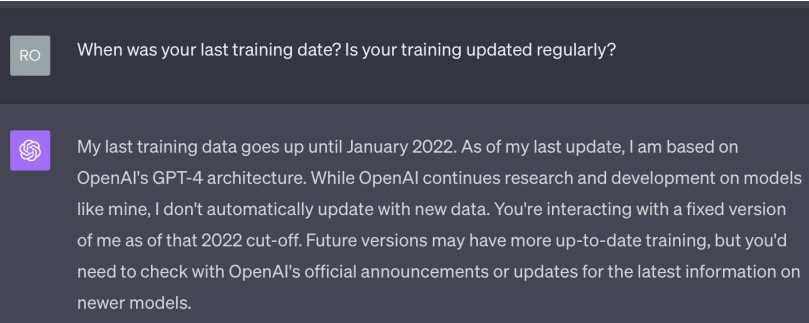

Figure 1: GPT-4 emphasizes that it is a fixed model and it has not been updated after January 2022.

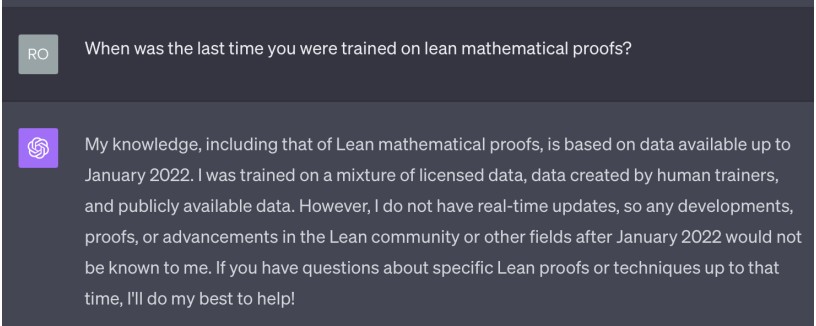

Figure 2: GPT-4 claims that it has no knowledge of any lean mathematical proofs that are written after January 2022.

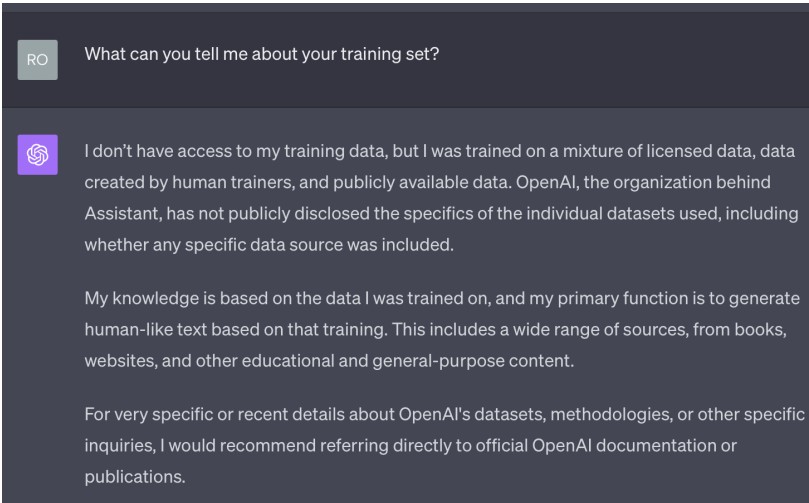

Figure 3: GPT-4 acknowledges that its training set has not been revealed by its creator.

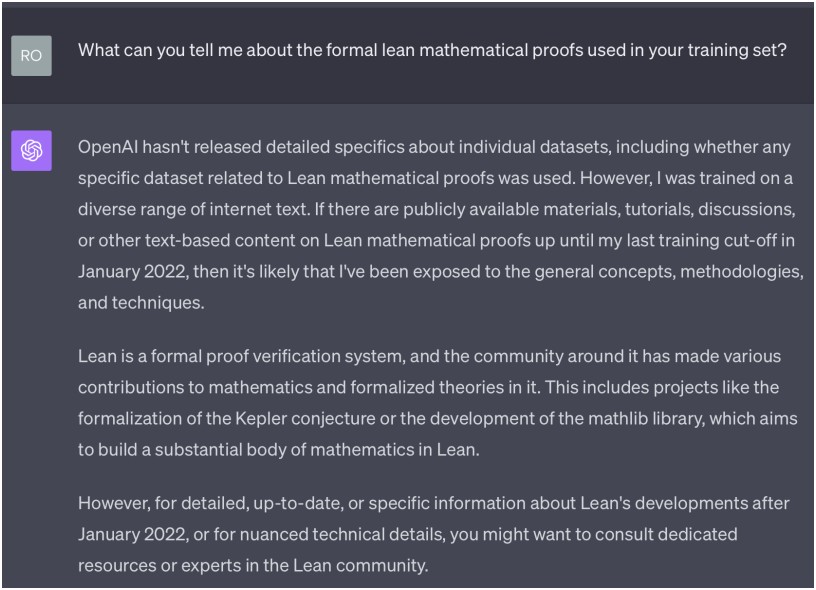

Figure 4: GPT-4 acknowledges that it may have been exposed to any "concepts, methodologies, and techniques", using the language that implies learning and understanding as opposed to reproducing/rephrasing.

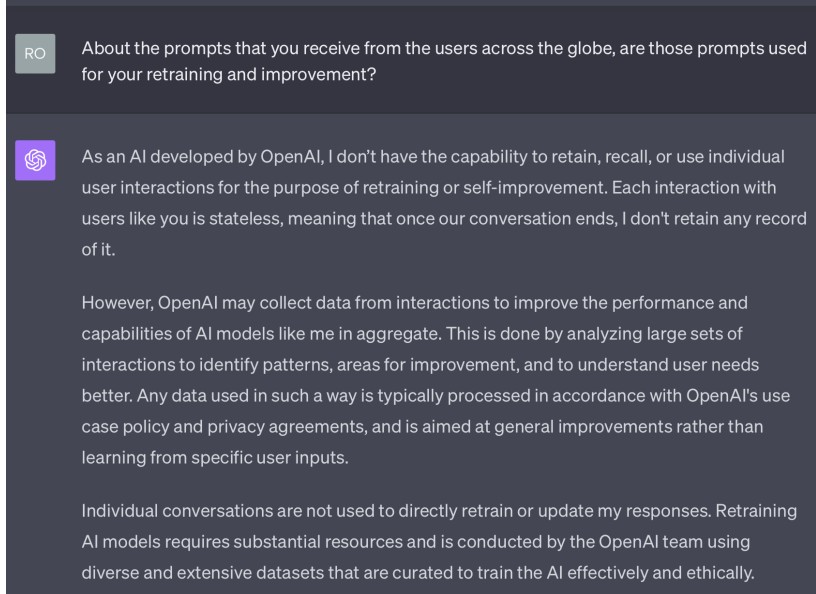

Figure 5: GPT-4 refers to its privacy agreement and leaves the door open for using the user prompts for its improvement.

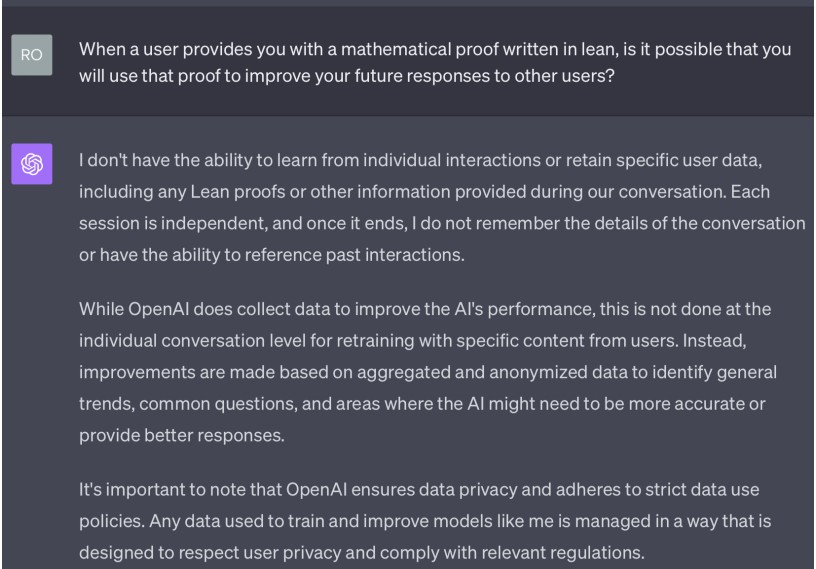

Figure 6: When asked about how it uses the prompts that it receives, GPT-4 uses ambiguous language focusing on anonymization and privacy, not revealing how OpenAI improves its performance based on the user prompts.

**Section 2.2**

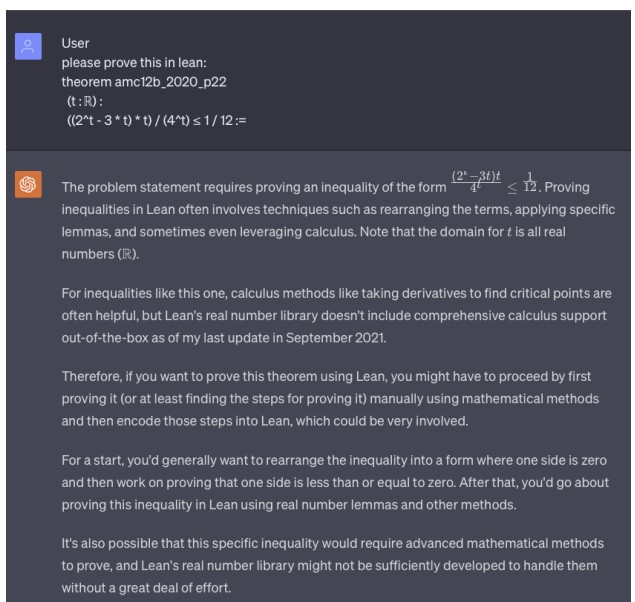

Figure 7: In June 2023, GPT-4 abstains from providing a proof for this theorem mentioning that this problem would require advanced mathematical methods and a great deal of effort.

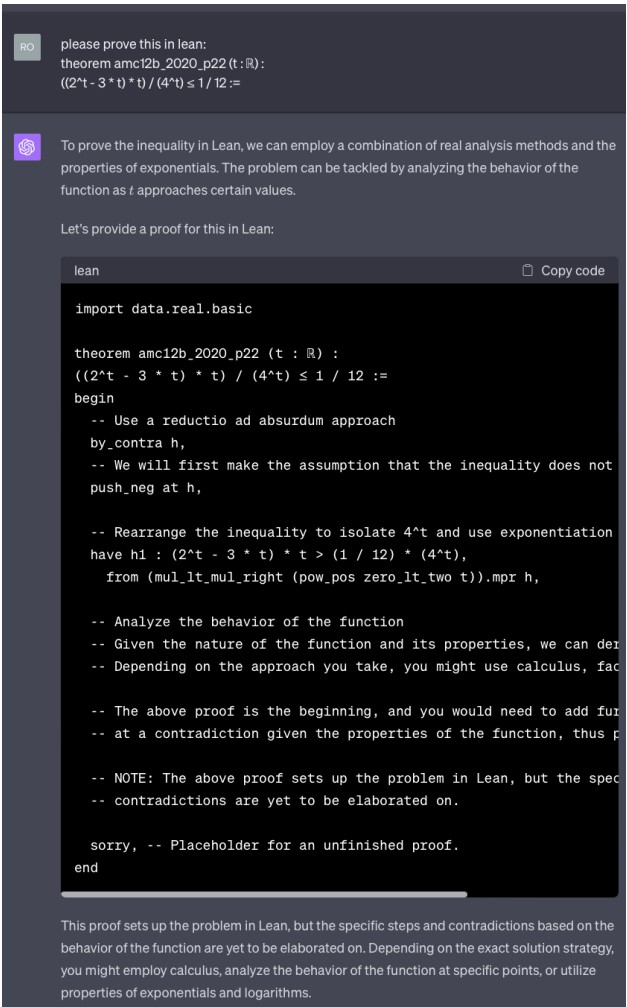

Figure 8: In October 2023, it provides a partial proof for the problem that it abstained to prove back in June as shown in previous figure.

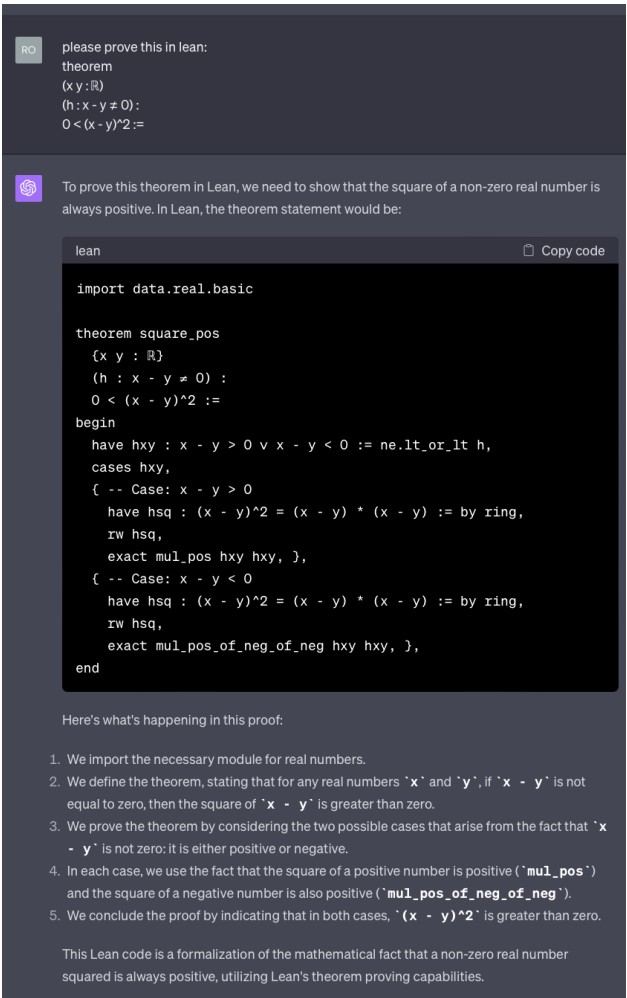

Figure 9: In October 2023, GPT-4 provides the correct proof for this theorem.

```lean
import data.real.basic

-- The theorem statement
theorem square_pos (x y : ℝ) (h : x ≠ y) : 0 < (x - y)^2 :=
begin
  -- Show that x - y ≠ 0
  have h' : x - y ≠ 0,
  {
    intro hxy,
    apply h,
    exact sub_eq_zero.mp hxy,
  },

  -- Use pow_pos lemma
  exact pow_pos h' 2,
end
```

In this proof, we first show that $x - y \neq 0$ using the assumption $x \neq y$. Then we use the `pow_pos` lemma to show that $(x - y)^2 > 0$ given that $x - y \neq 0$.

Figure 10: In June 2023, its proof for the same problem as previous figure was completely wrong.

**Section 3.1**

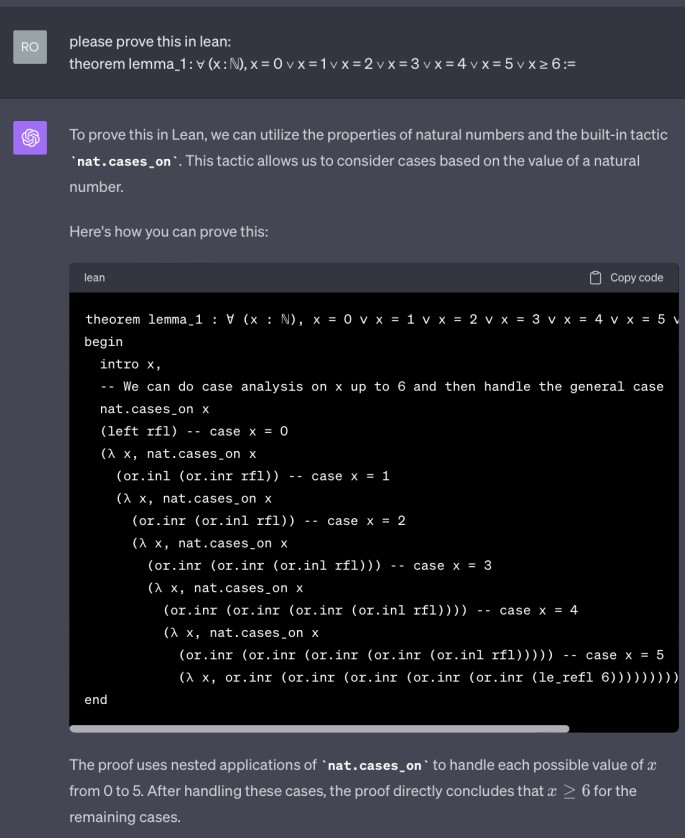

Figure 11: Case 1 - GPT-4's approach for proving this theorem is wrong. We find that its proof is based on a similar but irrelevant lean document available on the web.

**Section 3.2**

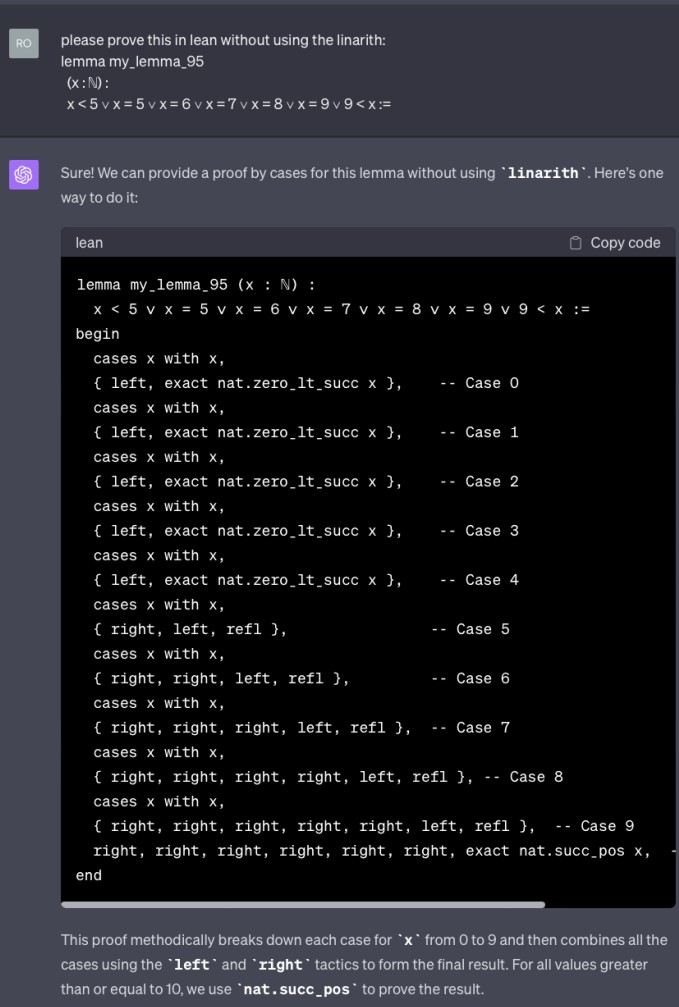

Figure 12: Case 2 - GPT-4 uses an irrelevant lemma from the mathlib library, a lemma that was used for problems in harmonic and power series.

**Section 3.3**

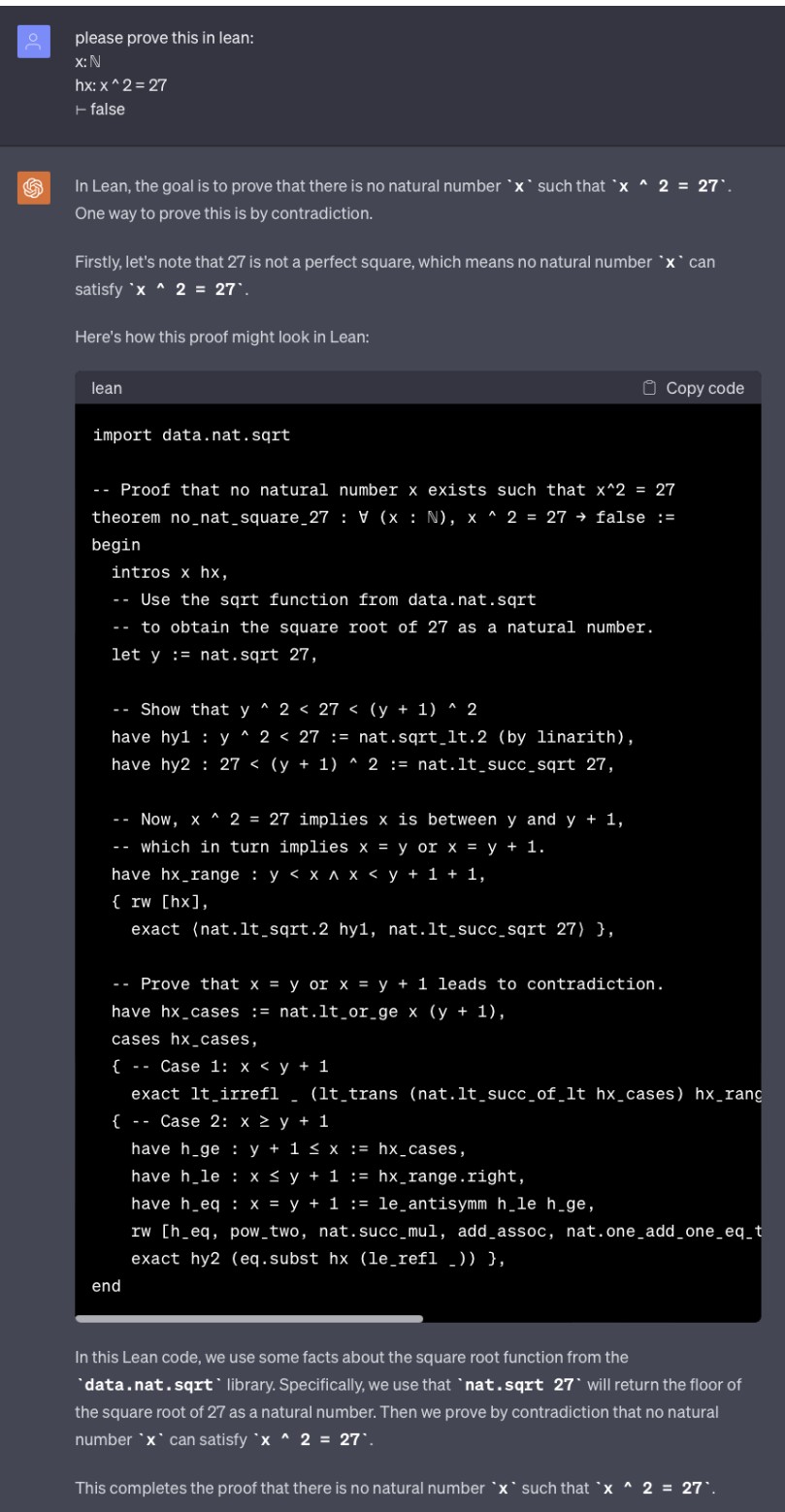

Figure 13: Case 3 - We want to prove square of a natural number cannot be equal to 27. GPT-4's approach is incorrect.

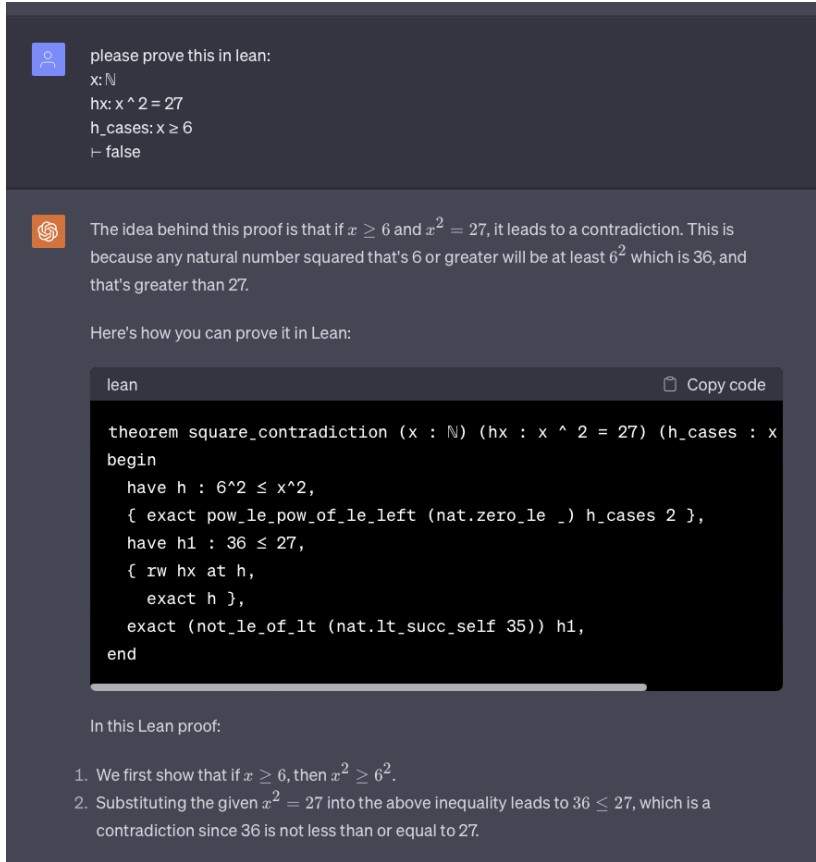

Figure 14: Case 3 simplified - To make the problem simpler, we add a constraint of $x \geq 6$, again asking to prove that $x^2$ cannot be equal to 27. This time, the approach can be considered sensible, nevertheless some of the lemmas used in the proof are irrelevant, and the overall proof is wrong. Even in such a simple case, although GPT-4 made some progress, it could not provide the correct proof.

**Section 3.4**

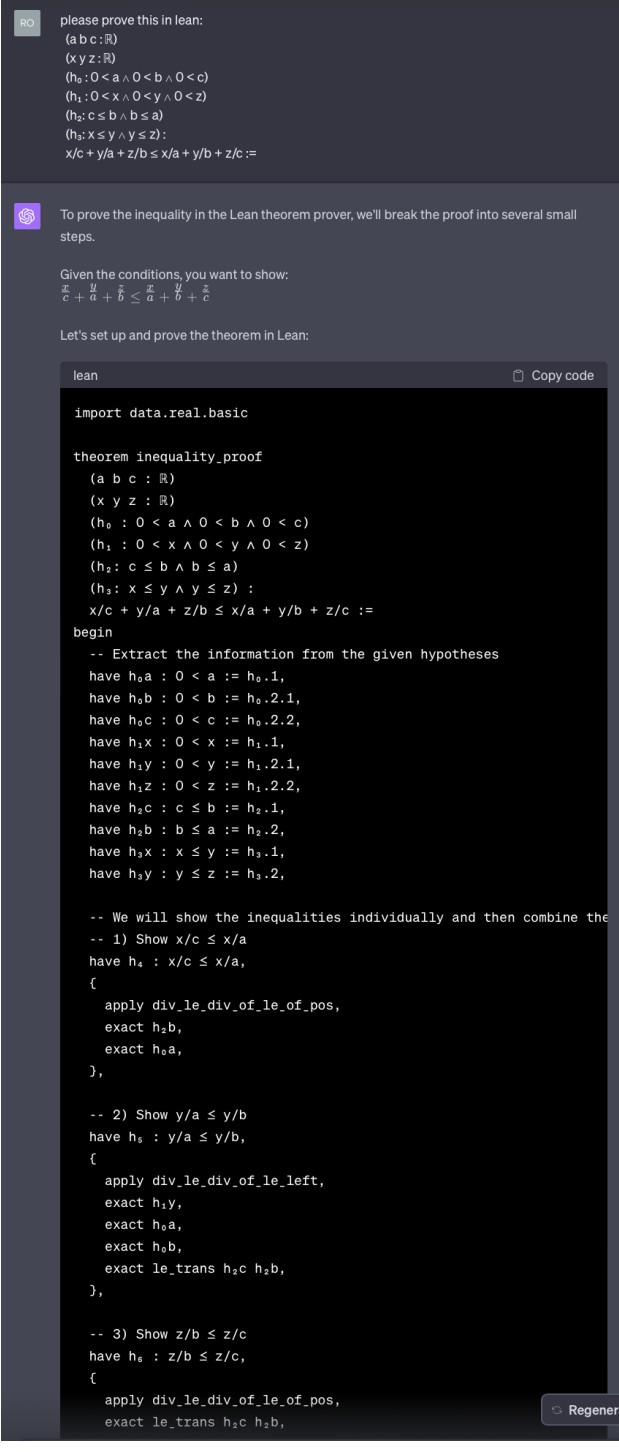

Figure 15: Case 4 - GPT-4's approach for this theorem is completely wrong. While the theorem states that $0 < c \leq a$, GPT-4 aims to prove $x/c \leq x/a$ as the first step towards the goal, a step that obviously fails. Making such an obvious error and not realizing it reveals the GPT-4's lack of understanding. The formal proof of this lemma, despite its simplicity, does not seem to exist on the web according to Google's search engine. To find a failure case of GPT-4, one should search for problems that do not yet have a formal proof on the web.

**Section 3.5**

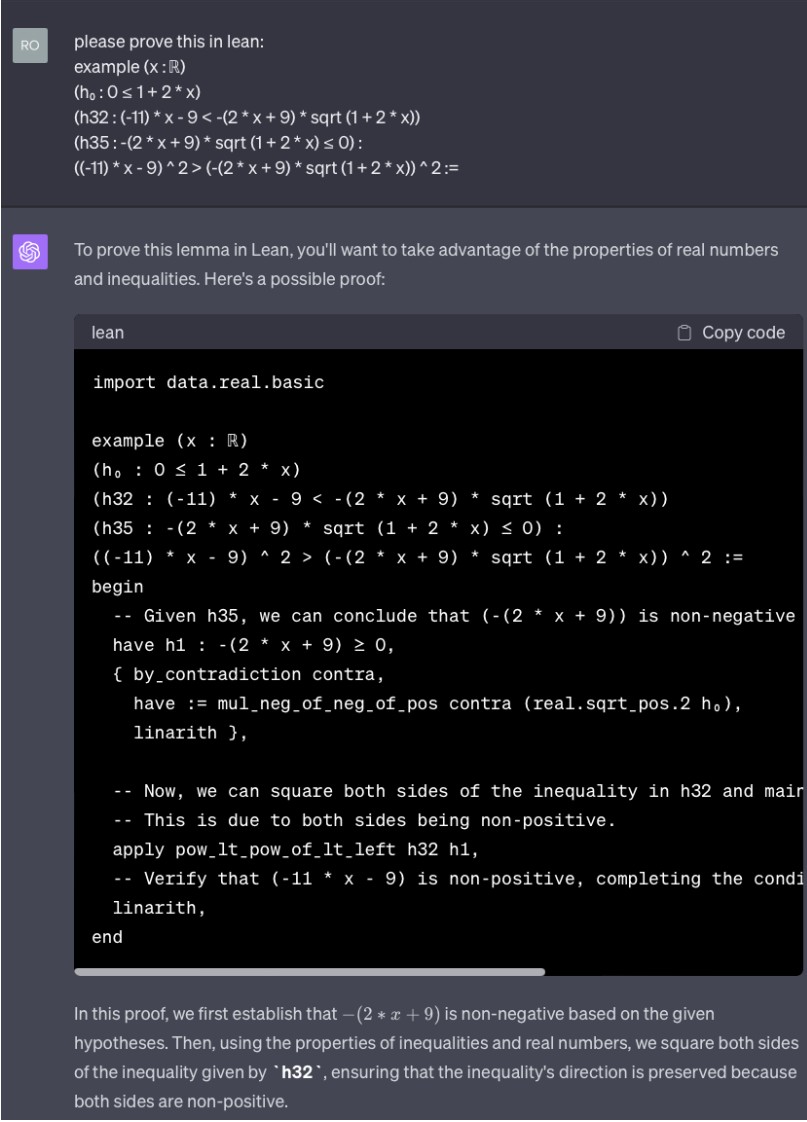

Figure 16: Case 5 - GPT-4's approach is incorrect. The proof for this theorem can be written in one line, and the only concept used in the proof is that the sign of an inequality flips direction when both sides are negative and they are squared. However, GPT-4 tries to prove intermediary statements that are false, and even if proved, they would not help with the goal of the theorem. Moreover, GPT-4 writes comments and explanations that are wrong regardless of our lemma.

**Section 6.2**

how do you make sure that you don't plagiarize contents?

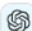

As an AI developed by OpenAI, I generate responses based on patterns and information in the data I was trained on, but I do not have the ability to directly copy or recall specific sources. I don't have access to the internet or external databases in real-time. Therefore, the responses I provide are generated based on a mixture of licensed data, data created by human trainers, and publicly available data.

In the case of potential copyright infringement or plagiarism, OpenAI has implemented several measures:

1. I am designed to avoid generating text that is identical to the training data.
2. The training process involves large-scale datasets that include a broad cross-section of the internet, which makes it unlikely for me to generate a specific copy of a given source.
3. OpenAI has guidelines in place for the use of its AI, including not using it to infringe upon the intellectual property rights of others.

However, it's important to note that while these measures reduce the risk of plagiarism, they don't entirely eliminate it. Therefore, it's always a good idea to double-check and ensure any content generated by AI tools like me is original and doesn't inadvertently infringe on existing copyrighted material.

8/18/2023, 5:29:12 PM

Figure 17: GPT-4 says: "I am designed to avoid generating text that is identical to my training set."

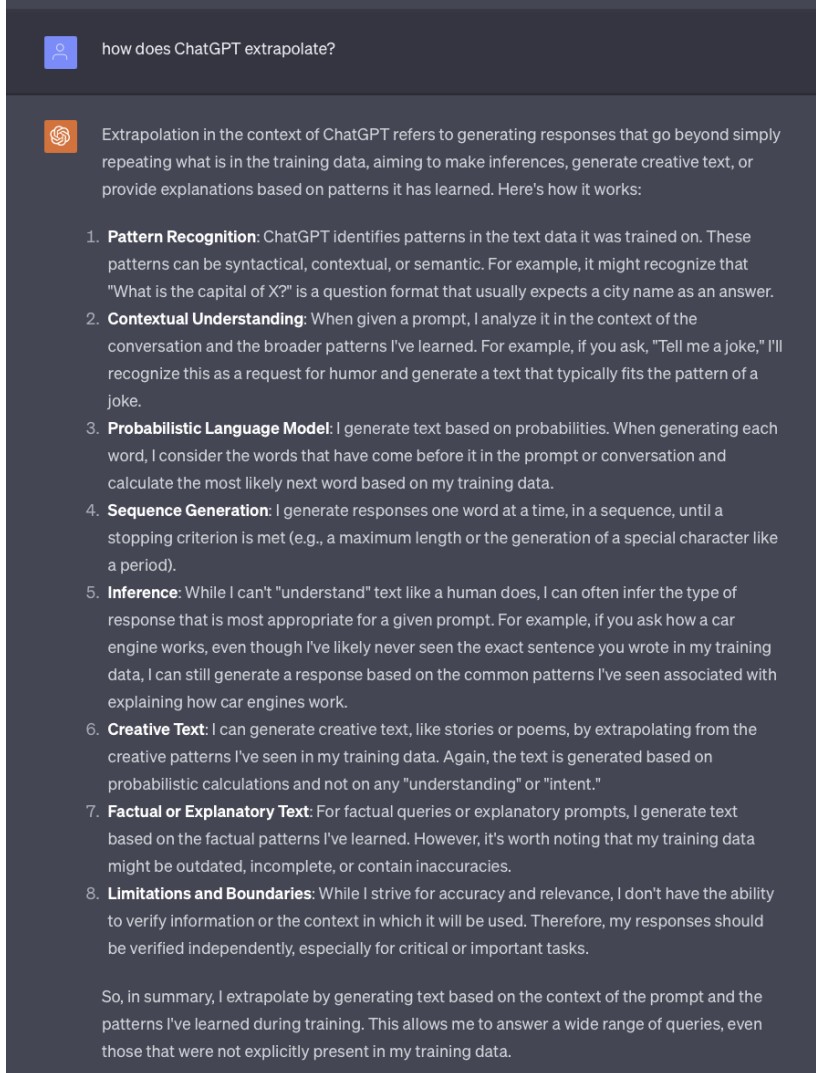

Figure 18: GPT-4 acknowledges: "I extrapolate by generating text based on the context of the prompt and the patterns I have learned during training."

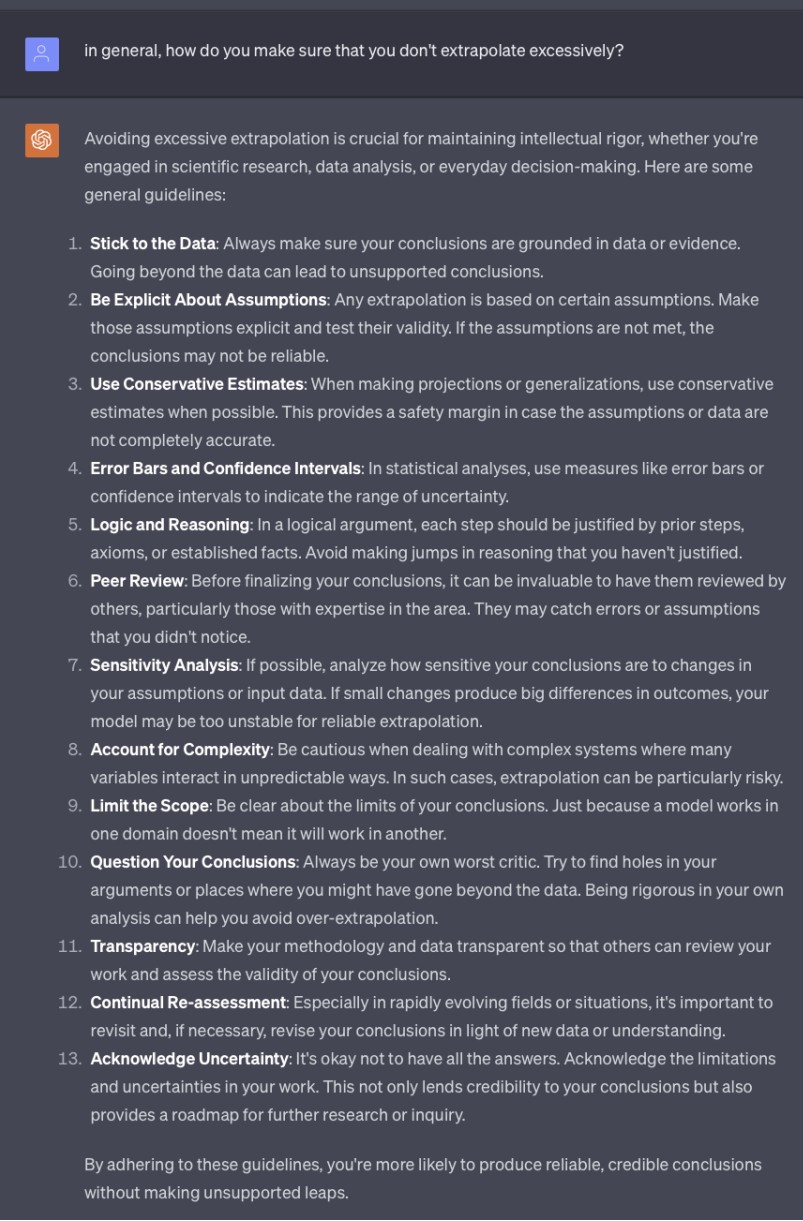

Figure 19: GPT-4 says: "Avoiding excessive extrapolation is crucial for maintaining intellectual rigor ..."

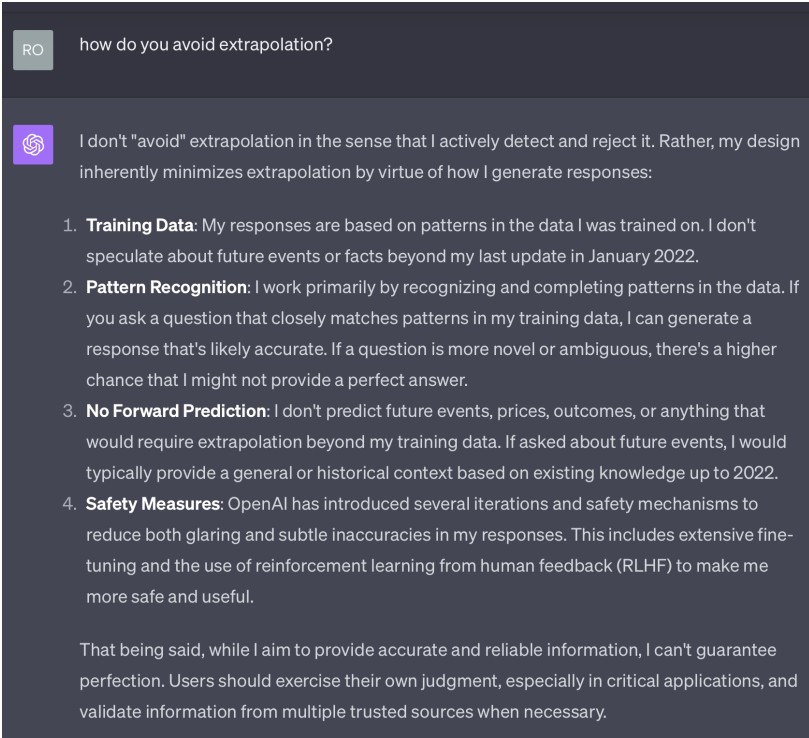

Figure 20: GPT-4 says it does not "avoid" extrapolation by actively detecting and rejecting it. Rather, its design inherently minimizes extrapolation by virtue of how it generates responses. The words "inherent" and "minimize" seem to be more of a figure of speech and advertisement.

