# OpenReview forum: "Large Language Models' Understanding of Mathematics: Source Criticism and Extrapolation"
_TMLR — Rejected by TMLR_

### Review · Reviewer_FxNb · 2023-11-29

**Summary Of Contributions:**

This paper contributes a case study on GPT-4's ability to prove lemma's in Lean and an opinion piece on multiple aspects of GPT-4, including its abilities as a mathematical theorem prover, a mathematical search engine, the way OpenAI communicates about GPT-4, and the accuracy of GPT-4's self-reporting. New knowledge that is presented comes in the form of a handful of prompts to GPT-4 and the analysis of these. Specifically, two prompts at two different points in time are used to claim that GPT-4 is a model that is changing over time, and six different prompts asking for proofs in Lean are used to claim GPT-4 does not understand basic mathematics. Most of the rest of the paper is an opinion piece about GPT-4 and OpenAI.

**Audience:**

No

**Broader Impact Concerns:**

I believe a broader impact statement is not necessary for the submission as it is presented currently.

**Claims And Evidence:**

No

**Requested Changes:**

Below I annotate each requested change with whether it's essential for an acceptance recommendation or whether it would not be essential but would strengthen the submission.

**Methodology changes**

- *Essential* -- My main issue with the case studies is that the confounding factor of the understanding of Lean is not discussed. I would recommend doing a study of GPT-4's grasp of the Lean programming language that is independent of it's mathematical understanding. E.g. can it spot mistakes in Lean? Can it fix errors in Lean, can it rewrite proofs in Lean?

- *Would strengthen the work* -- Consider using multiple phrasings of the same prompt to investigate variance in GPT-4's correctness, as well as prompting techniques that are known to increase reasoning capabilities of LLMs (zero-shot chain-of-thought, plan-and-solve, etc.).

- *Essential* -- You claim to have used many more examples to show the failure of GPT-4 when it comes to understanding math, but you picked 5 cases. I suggest you add these additional examples in an appendix, and report summary statistics on these in the main manuscript.

- *Essential* -- If you want to convincingly show that GPT-4 is a changing model over time a much larger empirical study needs to be conducted over multiple points in time than what is presented in this work.

**Writing changes**

- *Essential* -- Each claim in this paper needs to be either supported by citations, empirical or theoretical evidence, or weakened/removed.

- *Essential* -- The paper reads like an opinion piece, and the language can benefit from refraining from making strong unscientific claims. E.g. the first sentence of the paper and following it the entire introduction reads like a news article currently.

**Strengths And Weaknesses:**

### **Strengths**
- The authors recognise that to accurately evaluate the abilities of today's LLMs we need to design tasks that are not covered in the training set of the model, which is a difficult problem.
- The authors craft five tasks in the form of prompts that are unlikely to have been part of GPT-4's training data, but are still basic mathematic understanding tasks.

### **Weaknesses**

Most, if not all all, claims in the paper are not sufficiently or not at all supported by evidence. Before I go into detail, I will summarise my weaknesses here.
The main claims of the paper (i.e. GPT-4 does not understand basic mathematical concepts, is more like a faulty search engine, and is not a fixed model) are not sufficiently supported by evidence in the paper. Additionally, the methodology the authors use to evaluate GPT-4 has strong limitations (too few prompts and a confounding factor that could explain poor performance). Finally, the paper presents strong critiques about OpenAI and the way the company communicates, but does not provide evidence or citations for the things they claim OpenAI does.

Below, I will highlight two strong claims that are unsupported, and am happy to provide more detailed feedback on the other claims I am referring to above upon request.

### **The claim that GPT-4 can be used to rephrase proofs but does not grasp mathematical concepts**

The most important claim is already presented in the abstract, but also throughout the paper:
- *"Our finding suggests that GPT-4’s ability is to reproduce, rephrase, and polish the mathematical proofs that it has seen before, and not in grasping mathematical concepts."*
- *"Our results suggest that the task of theorem proving is more of a retrieval task such as Google’s search engine (Brin & Page, 1998) as opposed to the task of predicting the next word in a sentence."*

Notably, the authors do not provide empirical evidence of GPT-4's ability to *"reproduce, rephrase, and polish the mathematical proofs that it has seen before"*, and the rest of the claims about it's inability to understand are based on six zero-shot prompts. Besides the fact that it's difficult to claim a model does not grasp math based on such limited evidence, there is confounding factor in these tasks that is not investigated by the authors; namely the model's understanding of the programming language Lean. Perhaps the failure is a failure of using Lean and not mathematical understanding. Additionally, no prompting techniques are used that are known to aid model's reasoning abilities.

### **The claim that GPT-4 is not a fixed model**

Another claim the authors make is that GPT-4 is a changing model over time. While this is intuitively very likely to be true, the evidence provided in the paper is unsatisfactory. The authors prompt GPT-4 twice with two different prompts at different points in time, and show that GPT-4 gives an incorrect answer at first and a correct answer later. The authors use this evidence to claim that: *"We also see that GPT-4’s ability to prove mathematical theorems is continuously expanding over time despite the claim that it is a fixed model."* and that *"It is clear that GPT-4 is not just a single module model that predicts the next word, but on top of that, there is a sophisticated system of modules that pre-process the prompts and regulate its outputs."*. Without access to more information about OpenAI's methods, the observed behaviour in response to the two prompts can be explained by many things, including using a sampling temperature. We cannot claim based on the presented evidence that GPT-4 is a system of modules that pre-process the prompts.

### **A note on using self-reporting of GPT-4 as an indication of its inner workings**

Throughout the paper, the authors use self-reporting of GPT-4 as a kind of truth that should have informative value about the inner workings of OpenAI and the model, and I think this is misguided. OpenAI itself has a warning message at the bottom of the API that says that *"ChatGPT can make mistakes. Consider checking important information."*. For example, the authors use Section 2.1 to show a completion by GPT-4 claiming it's a fixed model to say that *"Researchers in the field refer to this claim to emphasize the importance of their success in proving theorems with GPT-4."*., but there's no citation for the claim that researchers do this. In Section 2.2 the authors try to debunk GPT-4's claim about being a fixed model. In Section 6.2 the authors again use GPT-4's self-reporting to discuss whether or not GPT-4 is trying to prevent itself from extrapolating too much. A potential study on how often what GPT-4 claims about its own abilities and inner workings could be interesting, but that does not seem the main purpose of this work nor is any convincing and systematic empirical evidence provided of whether or not GPT-4 is fixed and whether or not it is extrapolating or trying to minimise extrapolation.

### **A point on writing style**

The authors use inconsistent writing when referring to "GPT-4" and "OpenAI" throughout the paper; sometimes they are referred to with "the GPT-4" and "the OpenAI" and sometimes as "GPT-4" and "OpenAI". I'd say only the latter is correct.

---

### Review · Reviewer_2nqV · 2023-12-05

**Summary Of Contributions:**

The paper investigates GPT-4s ability to understand and and reason about mathematical tasks.

Particularly, the authors prompt GPT-4 with simple mathematical tasks, in Lean format, for which they can be certain that they are not on the web and therefore (likely) not part of the training data.
From qualitative analysis of six case studies (one in section 2.2, rest in section 3) the authors find that (i) GPT-4 does change over time, (ii) GPT-4 appears to be unable to to find proofs for lemmas that are no on the web/in the training data, and (iii) GPT-4 appears to be good at paraphrasing and polishing proofs from the training data, but not at producing new proofs.

In subsequent sections, the paper suggests practices for using GPT-4 in theorem proofing, discussing whether retrieval or next-word-prediction is better suited to the task, and lastly the role of compression and extrapolation in GPT-4s reasoning.

**Audience:**

No

**Broader Impact Concerns:**

No statement is present, but none is required.

**Claims And Evidence:**

No

**Requested Changes:**

I can not recommend acceptance of the manuscript in the current state.
In order to do so I'd like to see the weaknesses mentioned above, listed as actionable items below, to be addressed. In reference to reviewer instructions I have marked adjustments I'd require for my support as [critical] and those that would further strengthen the paper, but are not as crucial, as [optional].

- [critical] Extensive quantitative evaluation, ideally also beyond GPT-4 or a discussion of why this is not needed. (At the time of writing this review, two versions of GPT-4, `gpt-4` and `gpt-4-1106-preview`, are available, allowing to conduct the experiments without multi-month wait times.)
  - In these experiments the impact of the prompt in lean format is discussed [critical] or, ideally, experimentally accounted for [optional].
- [critical] Description of how the examples were picked (and further experimental parameters and choices).
- [critical] A more structured discussion of related work.
- [optional] Ideally, better integration of sections 5 and 6 into the paper and support for those claims via experiments.

I believe that these tasks are doable, but the resulting paper will likely be significantly different from the current manuscript.

**Strengths And Weaknesses:**

The paper is presented in an essay-style rather than the typical technical report style common in the field. Thus, while well written for an essay, I found it hard to follow.
Overall, the paper is timely as interest in the capabilities of Large Language Models (LLMs) in general, and mathematical theorem proofing in particular is growing. The idea of investigating lemmas that have not been published on the internet is interesting.

However, overall, the paper lacks rigor, experimental evidence and structure. Specifically:
- The paper does not clearly state its claims and contributions. Many conclusion in the paper are stated as "may" or "could", which makes it hard to judge what exact claims are made.
- Only GPT-4 is investigated.
  - This makes the title a slight over-claim.
  - Further, from an experimental perspective, the availability of open-source models [1,2] which are forthcoming about the composition of their training set, would allow to further test the authors hypothesis and contrast GPT-4 with these models.
- While some related works are discussed, no structured discussion of related work is provided. Areas that likely should be discussed:
    - Tool/Retrieval Augmentation of LLMs, in particular the case of combining LLMs with theorem provers. A very cursory search reveals at least [3,4,5,6], but many more exist.
    - Math Benchmarks and Datasets (beyond miniF2F) [7,8,9].
    - Likely further work on source criticism on internet sources should be discussed, but I am not familiar with this field.
- Lack of experiments and experimental details:
  - The set of experiments is very small and all evaluation is qualitative. Ideally, the claims derived form these would be backed up by quantitative evaluation on a larger set of examples.
  - I might have missed it, but it is not clear to me how the authors picked their examples.
  -  This makes it hard to judge the validity of the authors claims. For example, are these examples cherry-picked, such that no proof can be found?
  - The impact of prompting and the lean encoding is not discussed, beyond "we directly give the problem statement in formal Lean language as opposed to the natural language. This way, possible mistakes of translating to formal language will not arise". While this justification makes sense, from the model responses it seems this guides the LLM towards producing proofs in Lean syntax, which may skew the results, i.e., the model might be able to to reasoning in natural language. Further, it has been shown that prompting techniques, i.e., Chain-of-Thought [10], can have a large impact on the output of LLMs.
- Directly asking GPT-4 about itself in sections 2.1 and 6.2, should not be treated beyond anecdotal evidence.
- While sections 1-4 tell a coherent story, sections 5 and 6 are not well integrated into the paper. Similarly, the biographical information about Andrey Kolmogorov in section 6, while a interesting read, does not seem relevant to the paper.


Some minor comments:
- Inconsistent style of quotation marks. The first 4 pairs of quotation marks are matched (all page 1), but the rest (also starting page 1) are not.
- Typo: Chapter 3, page 5 third paragraph says "mst" instead of "most".

### References
[1] Llama 2: Open Foundation and Fine-Tuned Chat Models, Touvron et al., arXiv:2307.09288, 2023
[2] Llemma: An Open Language Model For Mathematics, Azerbayev et al., arXiv:2310.10631, 2023
[3] LeanDojo: Theorem Proving with Retrieval-Augmented Language Models, Yang et al., NeurIPS'23
[4] Draft, Sketch, and Prove: Guiding Formal Theorem Provers with Informal Proofs, Jiang et al., ICLR'23
[5] Baldur: Whole-Proof Generation and Repair with Large Language Models, First et al., ESEC/FSE'23
[6] A Language-Agent Approach to Formal Theorem-Proving, Thakur et al., arXiv:2310.04353, 2023
[7] FIMO: A Challenge Formal Dataset for Automated Theorem Proving, Liu et al., arXiv:2309.04295, 2023
[8] ProofNet: Autoformalizing and Formally Proving Undergraduate-Level Mathematics, Azerbayev et al., arXiv:2302.12433, 2023
[9] TRIGO: Benchmarking Formal Mathematical Proof Reduction for Generative Language Models, Xiong et al., EMNLP'23
[10] Large Language Models are Zero-Shot Reasoners, Kojima et al., NeurIPS'22

---

### Review · Reviewer_b2Mu · 2024-01-12

**Summary Of Contributions:**

This paper addresses the topic of using GPT-4 for theorem proving with the Lean proof assistant. The authors hypothesize that GPT-4's ability to prove new theorems is weaker than some may have suggested. The paper discusses 2 theorems that GPT-4 failed to prove in June 2023 and correctly proved in October 2023., then 6 cases where GPT-4 produced an incorrect proof (at an unreported date). The paper concludes with a discussion of compression and extrapolation.

The paper reports 20 interactions with GPT-4 through ChatGPT:
- 6 asking about GPT-4 itself: its training set, when it was updated, and how user interactions are used to improve the model
- 4 showing how GPT-4 produced incorrect and correct proofs for two theorems at different times
- 6 incorrect proofs produced for 6 different theorems
- 4 about whether GPT-4 can plagiarize and extrapolate

**Audience:**

Yes

**Broader Impact Concerns:**

No concerns.

**Claims And Evidence:**

No

**Requested Changes:**

- Provide citations for "Nevertheless, when GPT-4 succeeds in solving [...] its success may be reported in papers to indicate scientific progress." and "Researchers in the field refer to this claim to emphasize the importance of their success in proving theorems
with GPT-4."
- Perform and provide the results of a systematic study, rather than relying on anecdotal evidence.
- Remove sections of the paper that rely on asking GPT-4 itself about GPT-4.

**Strengths And Weaknesses:**

Strengths:
- The paper addresses the topic of using LLMs for theorem proving, a timely topic with great technical depth.
- The paper grapples with the difficulties of evaluating LLMs when the datasets used may become part of the LLMs' training data, which is also another topic of great interest at the moment.

Weaknesses:
- Parts of the methodology relies on asking GPT-4 itself about how it works (for example, about its training set, extrapolation, and its propensity/capability to plagiarize), which is not a reliable method. LLMs do not have such introspective capabilities and would not be expected to have them unless they have been trained to specifically answer questions about themselves.
- The paper relies on anecdotal evidence, relaying three (partial) successes and five failures at the authors' attempts to use GPT-4 for proving theorems in Lean.
- The paper speculates that GPT-4 failed to prove two theorems in June 2023, but (partially) succeeded in October 2023, due to the correct proof having been added to GPT-4's training data or other databases that it may use. There is no way to check whether this is correct.
- The discussion about GPT-4's training set and compression/extrapolation is not supported by theory or empirical evidence.

---

### Author Response · Authors · 2024-01-24
**General response**

Dear Reviewers,

Thank you very much for your thoughtful comments and for your constructive feedback. We address the common points in the reviews here, and will be happy to answer any remaining issues individually.

Before addressing the main points raised in the reviews, we would like to make a brief clarification about our contribution. We aim to evaluate the ability of LLMs (as exemplified by GPT-4) on writing formal proofs for easy problems, and to evaluate whether that ability goes beyond the formal proofs already available on the web (web is used as a proxy for the training set).

LLMs can be used in many different ways when it comes to writing formal proofs. For example:
  1. as a tool to translate a proof from natural language to formal language [4]
  2. as a tool to assist a mathematician in writing proofs, e.g., suggesting lemmas [11]
  3. as a tool to generate proof steps that could be verified and used by other systems [12]
  4. as a tool to write end to end formal proofs for specific problems (spectrum of easy to hard problems)

We only address the last item and focus on the range of easy problems that can lead to proofs for IMO problems. In section 3 of our paper, we discuss why this deserves a specific focus. In short, we want to cover the basics.

---

> ### Author Response · Authors · 2024-01-24
>
> The following thread is our response to the points raised in the reviews:
>
> **The point about large scale and systematic empirical analysis:**
>
> To address this point, we are performing a set of experiments based on 7 IMO (International Math Olympiad) problems in the miniF2F testing set. We specifically chose 7 problems that did not have a formal proof in lean (to the best of our knowledge) [14,16]. We (as humans) first wrote the complete proofs for these problems in Lean leading to more than a thousand lines of lean code. We then broke down these proofs into their smallest pieces. These pieces are required to construct the complete proofs for our IMO problems, but each piece by itself can be considered easy to solve and sometimes even trivial. Here are some examples:
>
>  - y is a natural number and y^2=4, prove that y equals 2.
> - q and r are integers and their product is 11, prove that q is either +- 1 or +- 11
> - q and r are natural numbers and their product is 5, prove that either q or r must be 1.
> - prove that 19 is a prime number
> - p is a natural number greater than 3, prove that rational number p/(p-1) <= 4/3
> - k is a natural number greater than 5, prove that 4 divides 2^k
> - k is natural number greater than 4, prove that k < 2^(k-2)
> - x,y, and i are positive natural numbers and we have x^i = y^i, prove that x=y
>
> For each of these small lemmas, we build similar lemmas with small modifications. For example, for the lemma:
> - k is a natural number greater than 5, prove that 4 divides 2^k,
> We also experiment with various values of n, such as 10, 15, 20, 100:
> - k is a natural number greater than n, prove that 4 divides 2^k
>
> Our library of basic lean lemmas includes harder steps such as proving injectivity of functions, points about rational numbers, casting, divisibility, but such steps make up a small fraction of our library. The majority of our lemmas are easy and do not have a formal proof on the web and those are the ones that we are most interested in.
>
> The larger steps in the proofs include steps and techniques such as induction, making assumptions without loss of generality, etc. The proof for each of these larger steps breaks down into many small steps. We focus on the small steps, i.e., the building blocks.
>
> This provided us with a systematic way to come up with easy problems and evaluate whether GPT-4 has the ability to solve the tiny building blocks of relatively challenging math problems. Our empirical experiments, except for a small section in the appendix, do not evaluate the ability of GPT-4 in putting the small steps together. Our main focus is to evaluate whether GPT-4, as the LLM with best performance on miniF2F, can write the proof for easy/trivial lemmas required to solve IMO problems. We are interested to evaluate the breadth of knowledge of GPT-4 and compare it against what is available on the internet.
>
> These led to a few hundred easy problems. We pose these problems to GPT-4 and record the responses. We then mark each response with one or more of the following labels:
> - conceptually correct/wrong in natural language
> - conceptually correct/wrong in lean
> - whether natural language and lean proof match each other
> - Errors in the lean proof
>   - None
>   - Hallucinations of lemmas and methods
>   - Wrong approach
>   - Correct approach but wrong implementation
>   - Minor errors, e.g., semantics
>
> With these results, one would know the obstacles that GPT-4 faces for solving IMO problems in the miniF2F. The first step for solving those IMO problems in miniF2F is the ability to write the proofs for the basic lemmas. The second step would be to put those basic lemmas together.
>
> We will release our lean proofs for the IMO problems and also our library of small lemmas along with the proofs.

---

> > ### Author Response · Authors · 2024-01-24
> >
> > **Source criticism:**
> >
> > For each of the small problems in our dataset, we search for their proof on the web, and we report whether or not Google’s search engine can find a formal lean proof for the problem.
> >
> > Putting this together with the performance of GPT-4, we report the correlation between the availability of a formal proof on the internet and the correctness of GPt-4’s response.
> >
> > Source criticism is a known method for evaluating sources of data, and specifically, sources of data available on the internet. Note that most other LLM’s, including Llama 2 and Llemma are trained on all the relevant data available on the internet, i.e., data scrapped via Common Crawl.
> >
> > Most other methods also suffer from the issue of possible leakage and unclear training set. For example, [5], the reference suggested by one of the reviews, honestly states:
> >
> > *“It is worth noting that, as with any LLM-based work, there is the potential for proofs from the test set to have leaked into the LLM pretraining data. While the pretraining data for the Minerva LLM at the base of our models does not include the PISA dataset, it does contain code that may include some Isabelle/HOL proofs found in PISA. This should be kept in mind when interpreting the results.”*
> >
> > **Chain of Thought and guiding the GPT-4:**
> >
> > 1- For the problems that GPT-4 does not solve correctly, we perform additional experiments by explaining the mistake in the proof and asking the model to correct its proof. We perform 5 rounds of corrections and report whether the model can eventually furnish the correct proof or not. We label the last response of GPT.
> >
> > 2- We explain the problems to the model in addition to the formal theorem statements and report the effect of that on the GPT-4’s response.
> >
> > 3- For each of the IMO problems,
> >   - We follow the chain of lean steps, and ask the GPT-4 to prove the next step and report whether it can prove the next step given everything else that is proved prior to that step.
> >   - At each of the lean steps, we ask the GPT-4 to prove the ultimate goal, and report how it performs.
> >
> > 4- We experiment by adding extra statements in the hypothesis set of theorems and evaluate whether that affects the performance of the model.

---

> > > ### Author Response · Authors · 2024-01-24
> > >
> > > **The point about the difficulty of evaluating how GPT-4 operates:**
> > >
> > > We were hoping to make this point in the paper. And it seems that reviewers largely agree with us.
> > > - Reviewer b2Mu states: “There is no way to check whether this is correct.” when they discuss whether the sources of information of GPT-4 are constant.
> > > - Reviewer 2nqV states: “Directly asking GPT-4 about itself in sections 2.1 and 6.2, should not be treated beyond anecdotal evidence.”
> > > - Reviewer FxNb states: “the authors use self-reporting of GPT-4 as a kind of truth that should have informative value”, and finds this misguided.
> > >
> > > We agree with these points in principle. The conclusion we derive from these points is that using GPT-4 is not a proper approach to measure scientific progress when it comes to solving problems that their solutions are potentially in the training set of the model. This is for a variety of reasons including the points quoted above from the reviews. If GPT-4 is used for benchmarking, at the least, it would be proper to report how its correct responses correspond to the proofs available on the web. Similarly, for other LLM's, comparing their correct responses against the content of their training set would be a necessity, in our view.
> > >
> > > When LLMs are tasked to solve problems where the solution is not on the web [12], or when they assist in generating an algorithm better than the best available algorithm [15], one knows that the solution is not a reproduction of what the model has seen before.
> > >
> > >
> > >
> > > **The point about asking GPT-4 about itself:**
> > >
> > > We do not necessarily agree with what GPT-4 says about itself. In fact, we have pointed out in a few instances, in our paper, that these answers appear to be mere claims. Nevertheless, since ChatGPT is being used in the literature as a tool to solve problems, it seems fitting to include what the model claims about itself, at least in the appendix of one of the papers in the literature.
> > >
> > > To address the reviews, we will move a portion of sections 5 and 6 to the appendix. We will also expand our literature review.
> > >
> > > Moreover, some of the responses that GPT-4 gives about itself are the most up to date information. For example, in November 2023 after we submitted our paper to TMLR, GPT-4 had a major update as indicated by a press release from OpenAI. This update indicated that its new training set was extended to include information up to April 2023. This was immediately reflected in the GPT-4’s responses about itself while the GPT-4 webpage and documentation on the OpenAI website is not updated.
> > >
> > > The details that GPT-4 says about its policy model and the system of modules that regulate its outputs and preprocess its inputs match the documentation of GPT-4, though the responses of GPT-4 provide more details. Whether the GPT-4 documentation and/or its responses are correct or not is not directly verifiable and we will clarify that we are not treating those responses as truth.

---

> > > > ### Author Response · Authors · 2024-01-24
> > > >
> > > > [1] Llama 2: Open Foundation and Fine-Tuned Chat Models, Touvron et al., arXiv:2307.09288, 2023
> > > >
> > > > [2] Llemma: An Open Language Model For Mathematics, Azerbayev et al., arXiv:2310.10631, 2023
> > > >
> > > > [3] LeanDojo: Theorem Proving with Retrieval-Augmented Language Models, Yang et al., NeurIPS'23
> > > >
> > > > [4] Draft, Sketch, and Prove: Guiding Formal Theorem Provers with Informal Proofs, Jiang et al., ICLR'23
> > > >
> > > > [5] Baldur: Whole-Proof Generation and Repair with Large Language Models, First et al., ESEC/FSE'23
> > > >
> > > > [6] A Language-Agent Approach to Formal Theorem-Proving, Thakur et al., arXiv:2310.04353, 2023
> > > >
> > > > [7] FIMO: A Challenge Formal Dataset for Automated Theorem Proving, Liu et al., arXiv:2309.04295, 2023
> > > >
> > > > [8] ProofNet: Autoformalizing and Formally Proving Undergraduate-Level Mathematics, Azerbayev et al., arXiv:2302.12433, 2023
> > > >
> > > > [9] TRIGO: Benchmarking Formal Mathematical Proof Reduction for Generative Language Models, Xiong et al., EMNLP'23
> > > >
> > > > [10] Large Language Models are Zero-Shot Reasoners, Kojima et al., NeurIPS'22
> > > >
> > > > [11] Song, P., Yang, K. and Anandkumar, A., 2023, October. Towards Large Language Models as Copilots for Theorem Proving in Lean. In The 3rd Workshop on Mathematical Reasoning and AI at NeurIPS'23.
> > > >
> > > > [12] Trinh, T.H., Wu, Y., Le, Q.V., He, H. and Luong, T., 2024. Solving olympiad geometry without human demonstrations. Nature, 625(7995), pp.476-482.
> > > >
> > > > [14] https://github.com/openai/miniF2F
> > > >
> > > > [15] Romera-Paredes, B., Barekatain, M., Novikov, A., Balog, M., Kumar, M.P., Dupont, E., Ruiz, F.J., Ellenberg, J.S., Wang, P., Fawzi, O. and Kohli, P., 2023. Mathematical discoveries from program search with large language models. Nature, pp.1-3.
> > > >
> > > > [16] Lample, G., Lacroix, T., Lachaux, M.A., Rodriguez, A., Hayat, A., Lavril, T., Ebner, G. and Martinet, X., 2022. Hypertree proof search for neural theorem proving. Advances in Neural Information Processing Systems, 35, pp.26337-26349.
> > > >
> > > > [17] Chen, L., Zaharia, M. and Zou, J., 2023. How is ChatGPT's behavior changing over time?. arXiv preprint arXiv:2307.09009.

---

> > > > > ### Comment · Reviewer_FxNb · 2024-01-27
> > > > > **Summary**
> > > > >
> > > > > Although the experimental protocols that the authors propose in the rebuttal are much improved, addressing requested changes about too few prompts and using SotA prompting techniques, implementing these will require a complete rewrite and significant change of the current manuscript. As it stands, I do not know what the results are to the revised experimental protocol presented in the author rebuttal. That, coupled with the fact that my concerns regarding confounding factors and many unsupported claims surrounding GPT-4 and OpenAI remain unanswered, cause me to stick to my original review of this paper and recommend rejection.

---

> > ### Comment · Reviewer_FxNb · 2024-01-27
> > **Response to **The point about large scale and systematic empirical analysis****
> >
> > The above described experimental setup is much improved from the one described in the manuscript. I am unsure whether the authors are saying they did this in their initial manuscript or that they did these experiments after the reviewing cycle, but either way it means that the manuscript needs an additional round of reviews (through a re-submission of a revised manuscript). What the authors describe in the above does not map onto the understanding I get from reading the manuscript. Additionally, I still do not know what the actual results are to the experimental setup described above, most importantly disentangling the confounding factor of using the Lean programming language.

---

> ### Comment · Reviewer_FxNb · 2024-01-27
> **Responses **Source criticism** and **Chain of Thought and guiding the GPT-4****
>
> Response to: **Source criticism**
>
> The fact that LLaMa 2 and LLemma are trained on common crawl has no bearing on whether OpenAI has added additional sources of training data to their design of GPT-4. The point remains that, though it's an important pursuit to figure out to what extent LLMs rely on their training data to provide mathematical proofs, GPT-4 is probably not the best model to test this in a scientific setup given that there is no way to test claims. If the proofs are not available on the internet it will be more likely that they are not in GPT-4's training data, but any claims surrounding this should be caveated.
>
> Response to: **Chain of Thought and guiding the GPT-4**
>
> Again, this sounds better but I am unsure whether the authors are proposing this or already did this. One addition I would make to this is few-shot examples in the prompt.

---

> ### Comment · Reviewer_FxNb · 2024-01-27
> **Responses to **The point about the difficulty of evaluating how GPT-4 operates** and **The point about asking GPT-4 about itself****
>
> Response to: **The point about the difficulty of evaluating how GPT-4 operates**
>
> I agree that in some cases using GPT-4 is not a proper approach to measure scientific progress, however I disagree with the authors that what they provide in their manuscript is a source of evidence for this claim.
>
> Response to: **The point about asking GPT-4 about itself**
>
> I disagree with the claim that because chatgpt is used to solve problems we should take self-reporting as a relevant thing to investigate. These two things seem entirely unrelated to me. Even though ChatGPT might not do accurate self-reporting (which OpenAI probably tries to prevent since the inner workings are not public on purpose), it can still be used in several ways as a tool to solve problems.

---

### Decision · Action_Editor_EYqa · 2024-04-02

**Recommendation:** Reject

**Comment:**

As mentioned above, the reviewers had several serious concerns about the paper's soundness. The authors have proposed to do additional experiments that address the concerns; however, these changes would require a significant rewrite of the paper, necessitating another round of reviews. Given this, I am recommending rejection for now. The authors are welcome to submit a major revision of the paper with the new experimental results.

**Audience:**

The paper studies LLMs' ability to perform mathematical reasoning, a topic that should interest many members of the TMLR community.

**Claims And Evidence:**

The paper studies the important question of whether LLMs can perform mathematical reasoning. Unfortunately, the reviewers found the execution to have major flaws -- several key claims lack proper experimental validation, parts of the methodology (for example, asking GPT about itself) are unsound, and the discussion of related work also needs substantial improvement. See the discussion below for more details.

**Resubmission Of Major Revision:**

The authors may consider submitting a major revision at a later time.